# Robust and annotation-free analysis of alternative splicing across diverse cell types in mice

**Gonzalo Benegas[1], Jonathan Fischer[2], Yun S Song[3,4,5]\***

[1]Graduate Group in Computational Biology, University of California, Berkeley, Berkeley, United States; [2]Department of Biostatistics, University of Florida, Gainesville, United States; [3]Computer Science Division, University of California, Berkeley, Berkeley, United States; [4]Department of Statistics, University of California, Berkeley, Berkeley, United States; [5]Chan Zuckerberg Biohub, Berkeley, United States

**Abstract** Although alternative splicing is a fundamental and pervasive aspect of gene expression in higher eukaryotes, it is often omitted from single-cell studies due to quantification challenges inherent to commonly used short-read sequencing technologies. Here, we undertake the analysis of alternative splicing across numerous diverse murine cell types from two large-scale single-cell datasets—the *Tabula Muris* and BRAIN Initiative Cell Census Network—while accounting for under-studied technical artifacts and unannotated events. We find strong and general cell-type-specific alternative splicing, complementary to total gene expression but of similar discriminatory value, and identify a large volume of novel splicing events. We specifically highlight splicing variation across different cell types in primary motor cortex neurons, bone marrow B cells, and various epithelial cells, and we show that the implicated transcripts include many genes which do not display total expression differences. To elucidate the regulation of alternative splicing, we build a custom predictive model based on splicing factor activity, recovering several known interactions while generating new hypotheses, including potential regulatory roles for novel alternative splicing events in critical genes like *Khdrbs3* and *Rbfox1*. We make our results available using public interactive browsers to spur further exploration by the community.

**\*For correspondence:**
yss@berkeley.edu

**Competing interest:** The authors declare that no competing interests exist.

## Editor's evaluation

This paper presents a new method to study known and novel alternative splicing events at the single-cell level and perform differential analysis across cell types. The method addresses current challenges in the analysis of splicing in single cells related to technical variation and experimental biases. Performing one of the most comprehensive studies to date with data from different mice, this work expands the body of splicing events that potentially define individual cell types.

## Introduction

The past decade's advances in single-cell genomics have enabled the data-driven characterization of a wide variety of distinct cell populations. Despite affecting more than 90% of human pre-mRNAs (*Wang et al., 2008*), isoform-level variation in gene expression has often been ignored because of quantification difficulties when using data from popular short-read sequencing technologies such as 10x Genomics Chromium and Smart-seq2 (*Picelli et al., 2014*). Long-read single-cell technologies, which greatly simplify isoform quantification, are improving (*Byrne et al., 2017*; *Gupta et al., 2018*; *Volden and Vollmers, 2020*; *Lebrigand et al., 2020*; *Joglekar et al., 2021*), but remain more costly

**eLife digest** Cells are the basic building blocks of all living things. There are numerous types of cells, and each cell has its own machinery to fulfill a specialised role. Despite their different purposes, most cells contain the same instructions, stored as DNA, on how to assemble the proteins needed to perform their intended functions. Cell types often vary in the frequency that each gene is read, leading to different quantities of proteins produced.

Moreover, a process known as alternative splicing enables cells to build multiple proteins from the same gene. It works by joining fragments of a gene's code in various combinations. The resulting RNA sequences are molecular templates that cells use to assemble proteins.

Analysing these RNA sequences reveals which genes are switched on in different tissues of the body, and what proteins are being made. However, despite recent advancements, alternative splicing is rarely studied in single cells because of some sizeable technical challenges.

Benegas, Fischer and Song developed a computational toolkit designed to handle the unique challenges of analysing alternative splicing events in single cells. The analysis pipeline, called scQuint, was tested on two large datasets that capture cell-to-cell differences in the brain and other tissues of mice.

Nearly all the cell types studied exhibited clear differences in alternative splicing, such that cell types could be distinguished based on their splicing profiles. Intriguing patterns of splicing were highlighted in some immune cells and certain types of neurons. Across cell types, the genes with unique splicing patterns were often not the same as those with unique activity patterns, indicating that gene expression and alternative splicing are two complementary processes. New types of alternative splicing events were also identified. Benegas et al. also developed a statistical model to probe the roles of splicing regulators in different cell types.

In summary, the scQuint toolkit overcomes critical technical challenges typically encountered when analysing alternative splicing in single cells. It also reveals new insights about mechanisms of alternative splicing. The results are open access, made available using public interactive browsers, which should spur on other researchers to interrogate how alternative splicing differs in single cells.

and lower-throughput than their short-read counterparts. For these reasons and others, short-read datasets predominate and we must work with short reads to make use of the rich compendium of available data. In response, researchers have developed several computational methods to investigate splicing variation despite the sizable technical challenges inherent to this regime. A selection of these challenges and methods are summarized in the Appendix.

To complement single-cell gene expression atlases, we analyze alternative splicing in large single-cell RNA-seq (scRNA-seq) datasets from the *Tabula Muris* consortium (*Schaum et al., 2018*) and BRAIN Initiative Cell Census Network (BICCN) (*Yao et al., 2021*). These data span a broad range of mouse tissues and cell types, and remain largely unexplored at the level of transcript variation. During our initial analyses, we encountered pervasive coverage biases, a heretofore largely unappreciated mode of technical variation which greatly confounds biological variation across cell types. Unsatisfied with the performance of current methods when confronted by these biases, we implemented our own quantification, visualization, and testing pipeline, named scQuint (single-cell quantification of introns), which allowed us to continue our analyses in a robust, annotation-free, and computationally tractable manner. Parts of the scQuint pipeline are based on adaptations of the bulk RNA-seq alternative splicing analysis method LeafCutter (*Li et al., 2018*) to handle the unique challenges of scRNA-seq data. As we demonstrate in subsequent sections, our modifications in the quantification, statistical modeling, and optimization procedures lead to improved robustness, scalability, and calibration when working with data from single cells (*Figure 2—figure supplement 2*, also see Materials and methods).

Applying scQuint to these datasets, we find a strong signal of cell-type-specific alternative splicing and demonstrate that cell type can be accurately predicted given only splicing proportions. Moreover, our annotation-free approach enables us to detect a large quantity of cell-type-specific novel splicing events. In certain cell types, particularly the neuron subclasses, as many as 30% of differential splicing events that we detect are novel. In general, across the many considered cell types and tissues in both datasets, we find only a narrow overlap between the top differentially expressed and the top differentially spliced genes within a given cell type, illustrating the complementarity of splicing

to expression. Our examination of neurons in the primary motor cortex suggests that splicing distinguishes neuron classes and subclasses as readily as does expression. We showcase alternative splicing patterns specific to the GABAergic (inhibitory) and Glutamatergic (excitatory) neuron classes as well as the subclasses therein. The implicated transcripts include key synaptic molecules and genes which do not display expression differences across subclasses. In developing marrow B cells, we find alternative splicing and novel transcription start sites (TSS) in critical transcription factors such as *Smarca4* and *Foxp1*, while further investigation reveals dissimilar trajectories for expression and alternative splicing in numerous genes across B cell developmental stages. These findings buttress our belief in the complementary nature of these processes and provide clues to the regulatory architecture controlling the early B cell life cycle. To facilitate easy exploration of these datasets and our results, we make available several interactive browsers as a resource for the genomics community.

Finally, to advance our understanding of alternative splicing regulation, we build a statistical machine learning model to predict splicing events by leveraging both the expression levels and splicing patterns of splicing factors across cell types. This model recovers several known regulatory interactions such as the repression of splice site four exons in neurexins by *Khdrbs3*, while generating new hypotheses for experimental follow-up. For example, in addition to the regulatory effect of the whole-gene *Khdrbs3* expression, the model predicts a regulatory role for a novel alternative TSS in this gene. In aggregate, our results imply that alternative splicing serves as a complementary rather than redundant component of transcriptional regulation and supports the mining of large-scale single-cell transcriptomic data via careful modeling to generate hypothetical regulatory roles for splicing events.

## Results
### Methods overview
### Robust, annotation-free quantification based on alternative introns
Most methods rely on the assumption that coverage depth across a transcript is essentially uniform (e.g., *Akr1r1*, **Figure 1—figure supplement 1a**). We instead found that Smart-seq2 data (**Picelli et al., 2014**) frequently contain sizable fractions of genes with coverage that decays with increasing distance from the 3' ends of transcripts. For example, in mammary gland basal cells from the *Tabula Muris* dataset (**Schaum et al., 2018**),

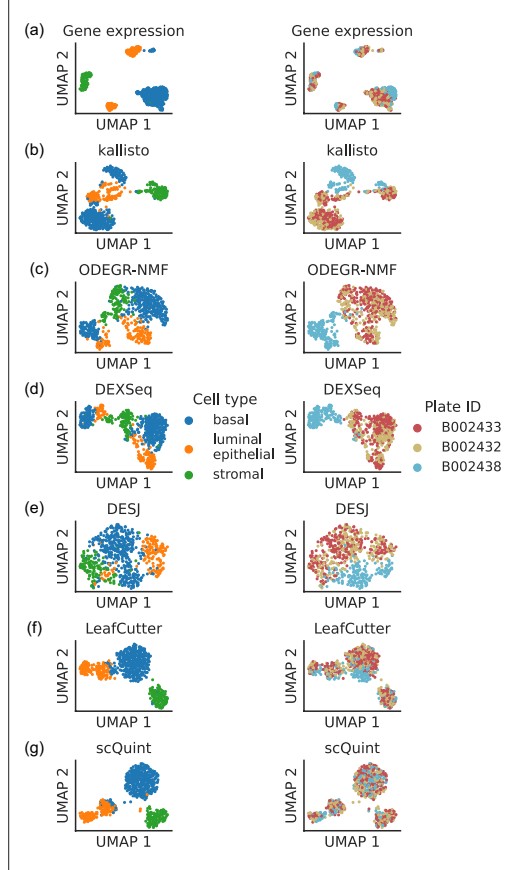

**Figure 1.** Clustering patterns by cell type and plate in the mammary gland from a three month-old female mouse in Tabula Muris. Cell embeddings based on different features were obtained by running PCA (gene expression) or VAE (the rest) followed by UMAP and subsequently colored by cell type (left column) and the plate in which they were processed (right column). (**a**) Gene expression, quantified using featureCounts (log-transformed normalized counts). (**b**) Isoform proportions. Isoform expression was estimated with kallisto and divided by the total expression of the corresponding gene to obtain isoform proportions. (**c**) Coverage proportions of 100 base-pair bins along the gene, as proposed by ODEGR-NMF. (**d**) Exon proportions, as proposed by DEXSeq. (**e**) Intron proportions across the whole gene, as proposed by DESJ. (**f**) Alternative intron proportions quantified by LeafCutter. (**g**) Alternative intron proportions (for introns sharing a 3' acceptor site) as quantified by scQuint.

The online version of this article includes the following figure supplement(s) for figure 1:

**Figure supplement 1.** Coverage artifacts in mammary gland basal cells from *Tabula Muris*.

**Figure supplement 2.** Technical artifacts in *BICCN Cortex*.

*Ctnbb1* shows a gradual drop in coverage (*Figure 1—figure supplement 1b*) while *Pdpn* displays an abrupt reduction halfway through the 3' UTR (*Figure 1—figure supplement 1c*). That the magnitude of these effects varies across technical replicates (plates) suggests they could be artifacts, possibly related to degradation or interrupted reverse transcription. Similar coverage bias artifacts are also apparent in the BICCN primary motor cortex data (*Yao et al., 2021*; *Figure 1—figure supplement 2*).

Such coverage biases affect gene expression quantification, and in some cases these batch effects are sufficient to comprise a significant proportion of the observed variation in expression levels. For the *Tabula Muris* mammary gland dataset, a low-dimensional embedding of cells based on gene expression reveals that some cell type clusters exhibit internal stratification by plate (*Figure 1a*). A subsequent test of differential gene expression between plate B002438 and all other plates returns 2870 significant hits after correction for multiple hypothesis testing, and all manually inspected differentially expressed genes exhibit these types of coverage biases. Perhaps unsurprisingly, quantification at the transcript level is apt to be even more sensitive to these artifacts than gene-level quantification, especially if it is based on coverage differences across the whole length of the transcript. The UMAP embeddings of isoform proportions (kallisto by *Bray et al., 2016*), exon proportions (DEXSeq by *Anders et al., 2012*), 100 bp bin coverage proportions (ODEGR-NMF by *Matsumoto et al., 2020*) or junction usage proportions across the whole gene (DESJ by *Liu et al., 2021*) depict a plate clustering pattern which scrambles the anticipated cell type clusters (*Figure 1b–e*).

With these considerations in mind, we sought to quantify transcript variation in a fashion that would be more robust to coverage differences along the transcript. Although some bulk RNA-seq methods such as RSEM (*Li and Dewey, 2011*) can model positional bias, they do so globally rather than in the gene-specific manner we encounter. One potential approach is alternative intron quantification as performed by bulk RNA-seq methods MAJIQ (*Vaquero-Garcia et al., 2016*), JUM (*Wang and Rio, 2018*), and LeafCutter (*Li et al., 2018*). Promisingly, quantification via LeafCutter (*Figure 1f*) yields an embedding that displays less clustering by plate than the other approaches we tried. We therefore based scQuint's quantification approach on LeafCutter's, with the key difference of restricting to alternative introns which share a common 3' acceptor site (*Figure 2*). This results in alternative splicing events that are equidistant from the 3' end of transcripts and which are less affected by the coverage biases we observed in scRNA-seq data. The embedding of cells based on our quantification approach (*Figure 1g*) shows less clustering by plate than LeafCutter and other methods.

Another advantage of alternative intron quantification is the ability to easily discover novel alternative splicing events. Whereas short reads generally cannot be associated with specific transcript isoforms, nor even exons if they partially overlap, split reads uniquely associate with a particular intron. Consequently, intron-based quantification does not depend on annotated transcriptome references and permits the discovery of novel alternative splicing events. This is important since, as detailed later, we estimate up to 30% of cell-type-specific differential splicing events are novel. Other annotation-free methods have been applied to single-cell short-read full-length data, but they do not provide a statistical test for differential splicing between two groups of cells (*Appendix 1—table 1*).

We do not recommend using scQuint to analyze alternative splicing in 10x Genomics Chromium data given its strong 3' transcript bias and evidence suggesting that these data can detect about half the number of junctions detected by Smart-seq2 (*Wang et al., 2021*). This imposes a fundamental limit on the number of transcripts that can be distinguished, and we expect alternative intron quantification to be sub-optimal in this setting. Nonetheless, several approaches for differential transcript usage in 10x data have been developed: Sierra (*Patrick et al., 2020*), SpliZ (*Olivieri et al., 2020*), and a kallisto-based approach which could be adapted for this task (*Ntranos et al., 2019*).

## Dimensionality reduction with Variational Autoencoder

To perform dimensionality reduction using splicing profiles, we developed a novel Variational Autoencoder (VAE) (*Kingma and Welling, 2014*) with a Dirichlet-Multinomial noise model, a natural distribution for sparse, overdispersed count data (*Figure 2b*, Materials and methods). For example, the often encountered 'binary' splicing (*Buen Abad Najar et al., 2020*) can be modeled by fitting a concentration parameter close to zero. VAEs are flexible and scalable generative models which have been successfully applied to analyze gene expression (*Lopez et al., 2020*) but have not yet been employed to investigate alternative splicing. To verify that we prevent leakage of gene expression information into our splicing profiles, we applied our VAE to embed a shuffled dataset obtained by resampling

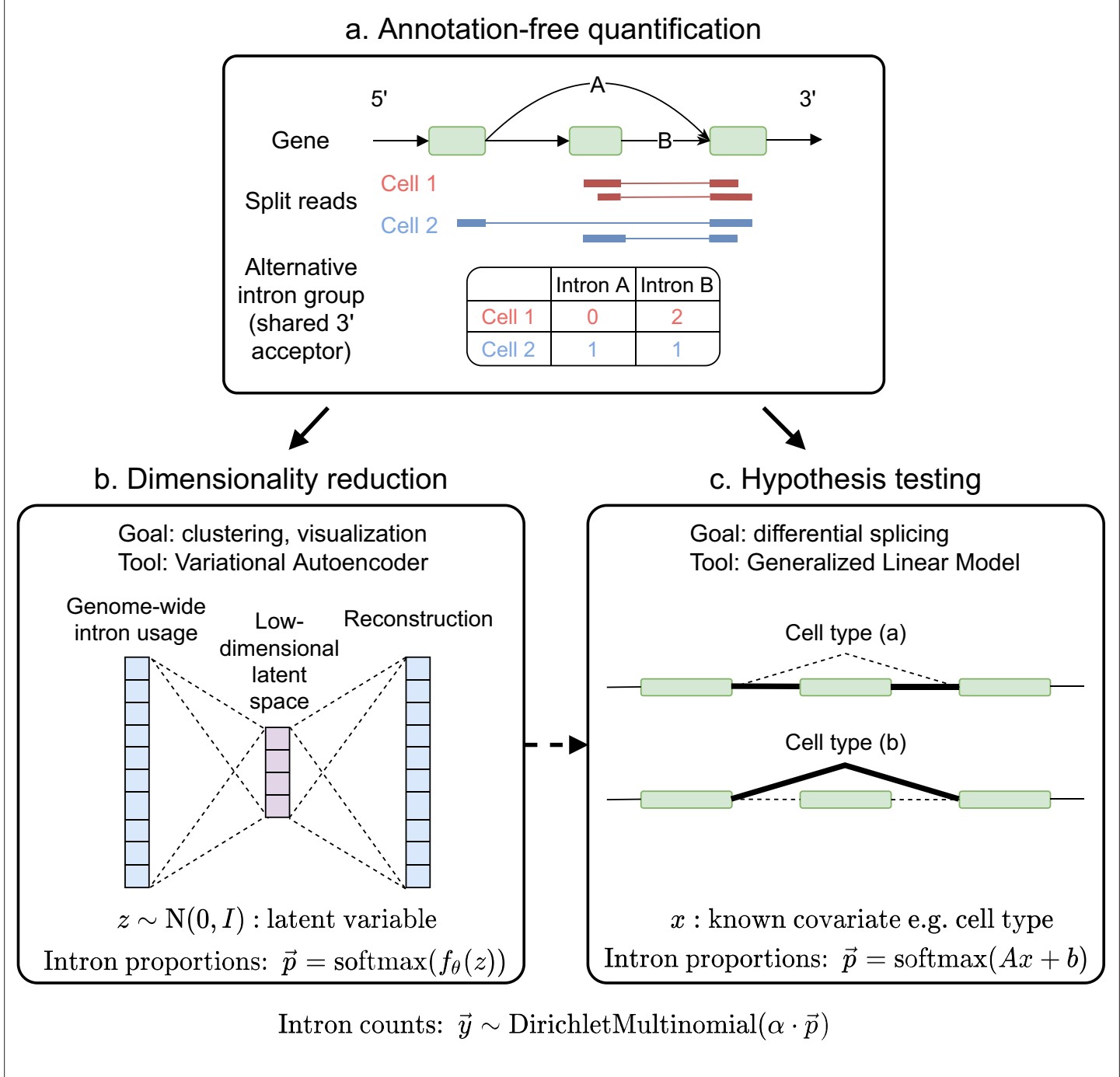

**Figure 2.** Overview of scQuint. (**a**) Intron usage is quantified from split reads in each cell, with introns sharing 3' splice sites forming alternative intron groups. (**b**) Genome-wide intron usage is mapped into a low dimensional latent space using a Dirichlet-Multinomial VAE. Visualization of the latent space is done via UMAP. (**c**) A Dirichlet-Multinomial GLM tests for differential splicing across conditions such as predefined cell types or clusters identified from the splicing latent space.

The online version of this article includes the following figure supplement(s) for figure 2:

**Figure supplement 1.** Splicing latent space when alternative intron counts are shuffled.

**Figure supplement 2.** Comparison with LeafCutter.

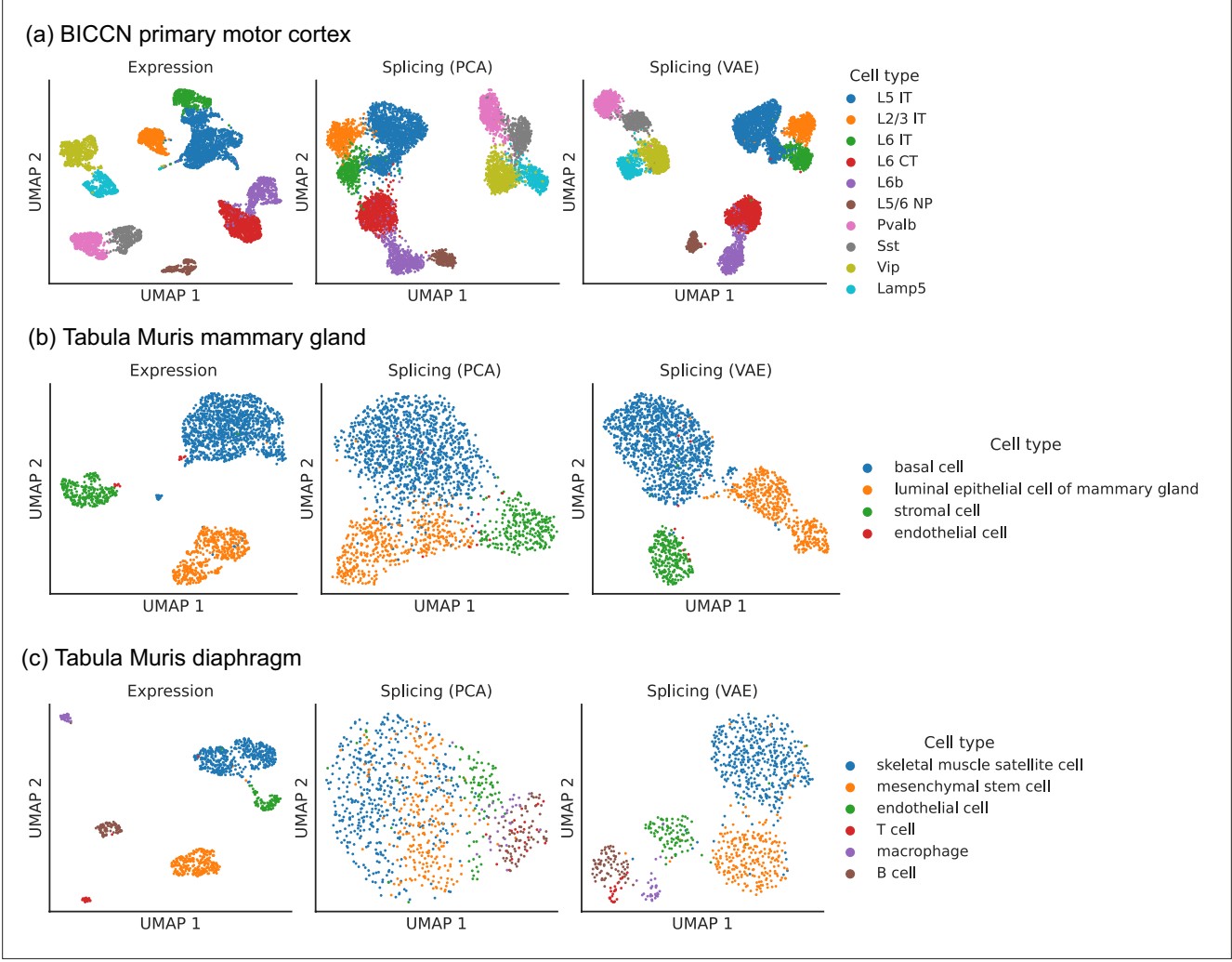

**Figure 3.** Comparison of splicing latent spaces obtained with PCA and VAE. Cells from (**a**) the cortex, (**b**) mammary gland and (**c**) diaphragm are projected into a latent space using PCA or VAE and visualized using UMAP. Cell type labels are obtained from the original data sources and are based on clustering in the expression latent space. The VAE is able to better distinguish cell types in the splicing latent space than PCA.

alternative intron counts with a fixed proportion in all cells. This shuffled dataset contained expression variability between cells but no splicing differences, and, as hoped, the resulting splicing latent space did not distinguish among cell types, indicating that it captures differences in splicing proportions rather than changes in absolute gene expression (*Figure 2—figure supplement 1*). We compared the latent space obtained with the VAE to the one obtained using Principal Component Analysis (PCA), a standard dimensionality reduction technique used in the LeafCutter and BRIE2 software packages. The VAE better distinguishes cell types than PCA (*Figure 3*), especially in the mammary gland and diaphragm.

## Differential splicing hypothesis testing with Generalized Linear Model

To test for differential splicing across cell types or conditions, we adopt a Dirichlet-Multinomial Generalized Linear Model (GLM) coupled with a likelihood-ratio test (*Figure 2c*, Materials and methods). We do so by adapting one of LeafCutter's proposed models for bulk RNA-seq to the scRNA-seq setting and apply it to our Smart-seq2 intron quantification. Namely, due to the sparse nature of scRNA-seq splicing data, we implement a more parsimonious statistical model featuring gene-level rather than intron-level parameters. Furthermore, we adjust the model-fitting algorithm at the initialization and optimization stages (see Materials and methods). After our modifications, we obtain well-calibrated p-values whereas those from LeafCutter's original differential splicing model are anti-conservative

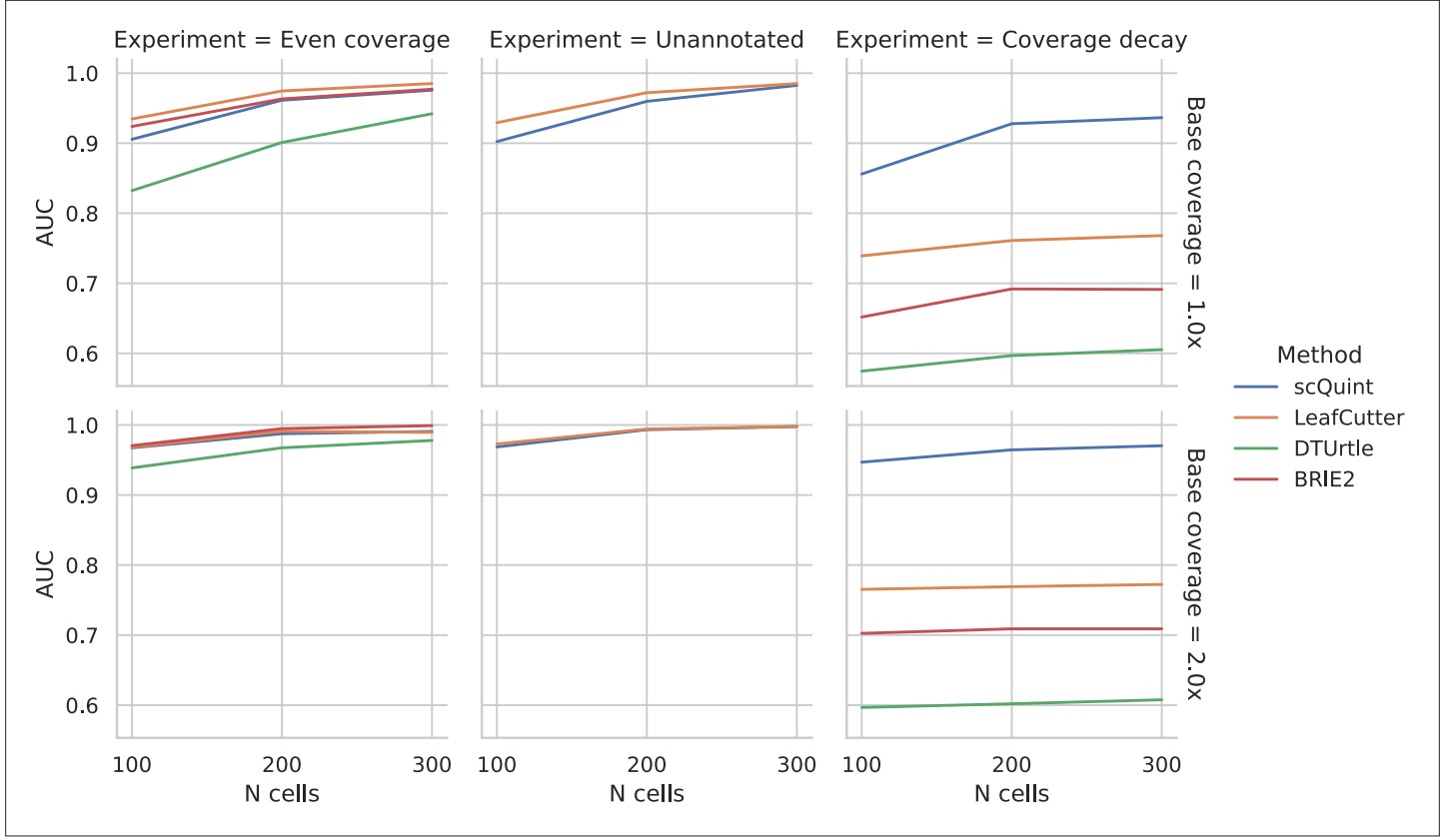

**Figure 4.** Evaluation of differential splicing test on simulated data. ROC AUC for detecting differential transcript usage between two groups, based on the p-value produced by different methods. *Unannotated*: the transcript reference given to methods is masked. *Coverage decay*: coverage decay with distance to the 3′ end of the transcript is induced in one of the two groups.

(*Figure 2—figure supplement 2*) and perhaps prone to extra false positives if applied directly to scRNA-seq data. We also find improvements in computational cost, both in runtime and memory usage (*Figure 2—figure supplement 2*).

As described in Materials and methods, we generated synthetic data in order to benchmark scQuint against three other methods that also offer two-sample tests for differential transcript usage proportions: BRIE2 and DTUrtle, both designed for scRNA-seq, and LeafCutter, designed for bulk RNA-seq (*Figure 4*). While the choice of an appropriate simulation model for scRNA-seq data is very much an open area of debate, particularly at the transcript level, we attempted to recreate a challenging setting for inference by assuming low coverage (1–2X) and high overdispersion (variance-to-mean ratio of 8). We performed three in silico experiments to assess performance under the differing conditions of even transcript coverage, unannotated events, and coverage decay across the transcript. In the case of even coverage, scQuint, LeafCutter, and BRIE2 perform similarly and do a good job of correctly identifying events, while DTUrtle is slightly behind. scQuint does only slightly worse with low cell counts and low coverage, which is probably a trade-off for the robustness that comes from only using reads from junctions sharing 3′ acceptor sites. Next, we recreated the unannotated setting by masking the reference given to methods. Only scQuint and LeafCutter are able to perform differential transcript usage testing in this setting, and, as expected, they performed nearly identically to the annotated setting with even coverage. Lastly, we created a setting where transcript coverage decays with distance from the 3′ in one of the two groups, mirroring a pattern we often saw in the real data analyzed for this paper. Here, scQuint outperforms the other tested methods by a wide margin with performance improving at higher coverages, unlike other methods. These results validate that scQuint is robust to both incomplete annotations and coverage decay while only paying a modest penalty relative to other methods under ideal conditions (even coverage and annotated events).

**Table 1.** Overview of analyzed datasets.

Number of cells, tissues, cell types, individuals, detected genes, and detected alternative introns (including the percentage of introns that are not present in the Ensembl reference) for both data sources.

| Dataset | Cells | Tissues | Cell types | Individuals | Genes | Alt. introns | Unannotated |
|---|---|---|---|---|---|---|---|
| BICCN Cortex | 6220 | 1 | 11 | 45 | 26,488 | 39,357 | 29% |
| Tabula Muris | 44,518 | 23 | 117 | 8 | 27,348 | 29,965 | 25% |

The online version of this article includes the following source data for table 1:

**Source data 1.** Number of cells per cell type and donor in BICCN Cortex.

**Source data 2.** Number of cells per tissue and donor in Tabula Muris.

## Augmenting cell atlases with splicing information

We applied scQuint to two of the largest available Smart-seq2 datasets. The first comprehensively surveys the mouse primary motor cortex (*BICCN Cortex*) (*Yao et al., 2021*) while the second contains over 100 cell types distributed across 20 mouse organs (*Tabula Muris*) (*Schaum et al., 2018*; *Table 1*). We detect more alternative introns in *BICCN Cortex* neurons than in the entire broad range of cell types present in *Tabula Muris* (which includes neurons but in much smaller number). This observation comports with previous findings that the mammalian brain has exceptionally high levels of alternative splicing (*Yeo et al., 2004*). *Booeshaghi et al., 2021* analyzed *BICCN Cortex* at the transcript level, but focused on changes in absolute transcript expression rather than proportions. While the authors indirectly find some differences in transcript proportions by inspecting genes with no differential expression, this is not a systematic analysis of differential transcript usage. Meanwhile, only microglial cells in *Tabula Muris* (*Nip et al., 2020*) have been analyzed at the transcript level. (*Tabula Muris* also contains 10x Chromium data analyzed at the transcript level [*Patrick et al., 2020*]).

As a community resource, we provide complementary ways to interactively explore splicing patterns present in these datasets (*Figure 5*), available at (https://github.com/songlab-cal/scquint-analysis, *Benegas, 2021a*) with an accompanying tutorial video. The UCSC Genome Browser (*Kent et al., 2002*) permits exploration of alternative splicing events within genomic contexts such as amino acid sequence, conservation score, or protein binding sites, while allowing users to select different length scales for examination. We additionally leverage the cell×gene browser (*Megill et al., 2021*) (designed for gene expression analysis) to visualize alternative intron PSI (percent spliced-in, defined as the proportion of reads supporting an intron relative to the total in the intron group) via cell embeddings. Further, one can generate histograms to compare across different groups defined by cell type, gender, or even manually selected groups of cells. These tools remain under active development by the community, and we hope that both the genome- and cell-centric views will soon be integrated into one browser.

## Cell-type-specific splicing signal is strong and complementary to gene expression

### Primary motor cortex

We first explored the splicing latent space of *BICCN Cortex* cells by comparing it to the usual expression latent space (*Figure 6a*). Cells in the splicing latent space strongly cluster by cell type (annotated by *Yao et al., 2021* based on gene expression). A similar analysis was recently performed (*Feng et al., 2021*) on a different cortex subregion in which most, but not all, neuron subclasses could be distinguished based on splicing profiles (e.g., L6 CT and L6b could not be separated). However, the authors only considered annotated skipped exons, a subset of the events we quantify, and used a different dimensionality reduction technique.

*Figure 6b* (top left) highlights some differentially spliced genes between Glutamatergic and GABAergic neurons, including the glutamate metabotropic receptor *Grm5* as well as *Shisa9/Ckamp44*, which associates with AMPA ionotropic glutamate receptors (*von Engelhardt et al., 2010*). The expression pattern of these genes, meanwhile, does not readily distinguish the neuron classes (*Figure 6b*, top right). In *Pgm2*, a gene of the glycolysis pathway thought to be regulated in the

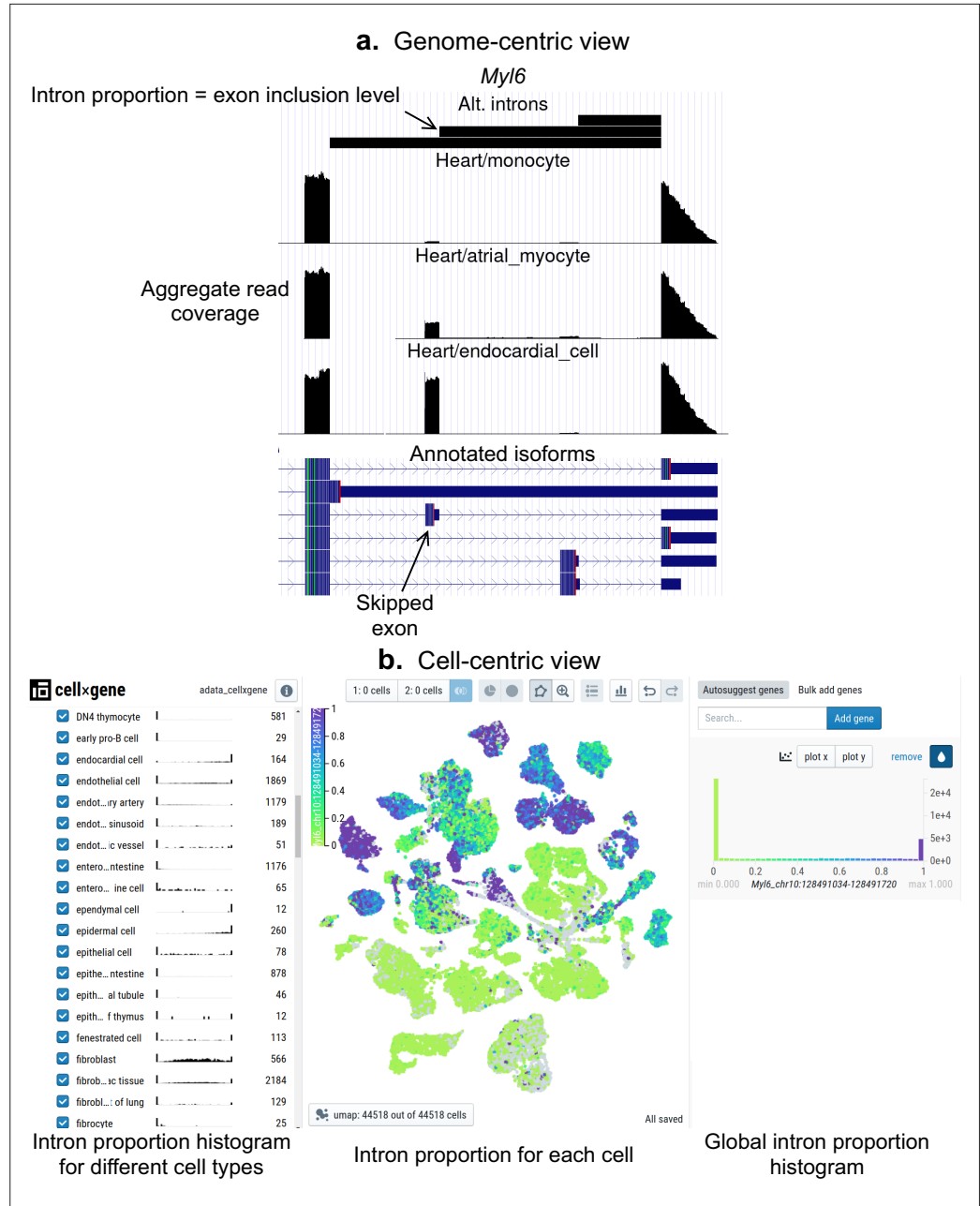

**Figure 5.** Interactive visualizations of splicing patterns. As an example, a skipped exon in *Myl6*. (**a**) The UCSC Genome browser visualization of this locus. Bottom: annotated isoforms of *Myl6*, including a skipped exon. Center: aggregate read coverage in three cell types with varying inclusion levels of the skipped exon. Top: three alternative introns that share a 3' acceptor site. The identified intron's proportion corresponds to the skipped exon's inclusion level. (**b**) cell×gene browser visualization of the marked intron's proportions (Myl6_chr10:128491034–128491720). Center: intron proportion for each cell in the UMAP expression embedding. Sides: intron proportion histogram for (left) different cell types and (right) all cells.

developing cortex by mTOR (*Schüle et al., 2021*), we discover a novel exon preferentially included in Glutamatergic neurons (*Figure 6c*, *Figure 6—figure supplement 2*).

Our differential splicing test reveals thousands of cell-type-specific splicing events (further discussed below in subsection Comparison of selected tissues), highlighting marker introns that distinguish neuron subclasses, while the expression of their respective genes does not; for example, compare the bottom left and bottom right panels of *Figure 6b*. Genes that better distinguish cell types at the expression level can be seen in *Figure 6—figure supplement 1*. As another example of the many

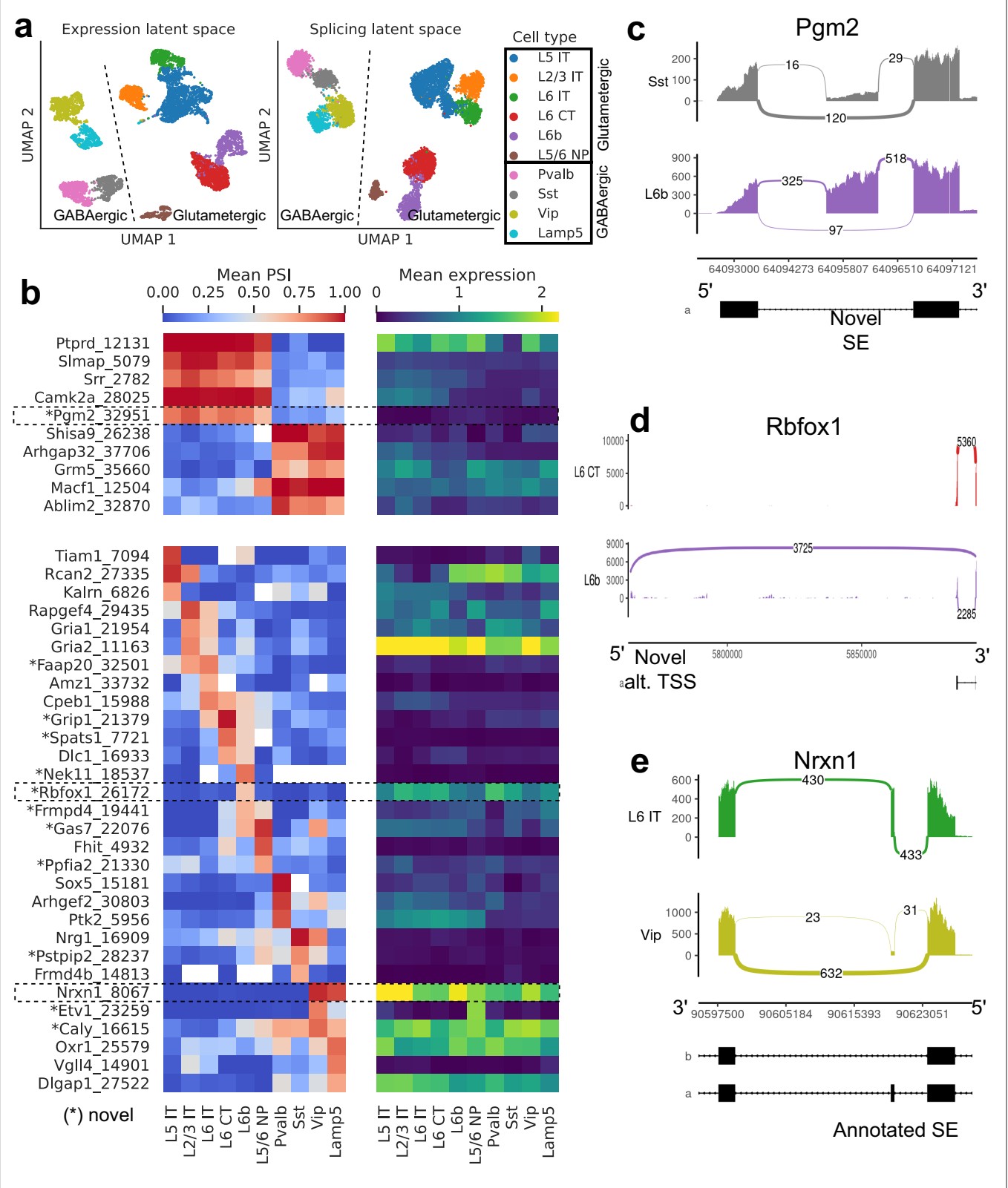

**Figure 6.** Splicing patterns in *BICCN Cortex*. (**a**) Expression and splicing latent spaces, visualized using UMAP. The expression (splicing) latent space is defined by running PCA (VAE) on the gene expression (alternative intron proportion, PSI) matrix. Cell types separate well in both latent spaces. (**b**) PSI of selected introns (left) and expression (log-transformed normalized counts) of their respective genes (right) averaged across cell types. Top: introns distinguishing Glutamatergic and GABAergic neuron classes. Bottom: introns distinguishing neuron subclasses. (**c–e**) Sashimi plots (***Garrido-Martín***

*Figure 6 continued on next page*

*Figure 6 continued*

*et al., 2018*) of specific alternative splicing events, displaying overall read coverage with arcs indicating usage of different introns (certain introns are shrunk for better visualization). (**c**) Novel skipped exon in *Pgm2*. (**d**) Novel alternative transcription start site (TSS) in *Rbfox1*. (**e**) Annotated skipped exon (SE) in *Nrxn1*.

The online version of this article includes the following source data and figure supplement(s) for figure 6:

**Source data 1.** Intron coordinates for panel (b).

**Figure supplement 1.** Marker genes for cell types in *BICCN Cortex*.

**Figure supplement 2.** PSI distribution of Pgm2_32951.

**Figure supplement 3.** PSI distribution of Rbfox1_26172.

**Figure supplement 4.** PSI distribution of Nrxn1_8067.

novel events we discover, we showcase a novel alternative transcription start site in *Rbfox1*, a splicing factor known to regulate cell-type-specific alternative splicing in the brain (*Wamsley et al., 2018*; *Figure 6d*, *Figure 6—figure supplement 3*). This novel TSS (exon chr16:5763871–5763913, intron Rbfox1_26172), which lies in a highly-conserved region, is (partially) used by only L6b neurons. We are also able to detect well-known cell-type-specific alternatively spliced genes such as *Nrxn1*, which encodes a key pre-synaptic molecule (*Figure 6e*, *Figure 6—figure supplement 4*; *Fuccillo et al., 2015*). In this case, we observe an exon (known as splice site 2) exclusively skipped in Vip and Lamp5 neurons.

## General patterns in *Tabula Muris*

We next turned our attention to *Tabula Muris*, which comprises a wide variety of organs and cell types from across the entire body. As before, we initially compared the expression and splicing latent spaces using UMAP (*Figure 7a*). This revealed broadly consistent clusters between projections, but a visible shift in the global layout of these clusters. In particular, whereas cell types were better separated in the expression projection, cell classes (e.g., endothelial, epithelial, immune) formed more coherent clusters in the splicing projection.

To supplement our qualitative comparison of UMAP projections with a more rigorous approach, we built dendrograms and a tanglegram using the respective distances between cells in each of the expression and splicing latent spaces (*Figure 7b*). Despite minor shifts, the dendrograms resemble one another, and most subtree structure is preserved. The low value of their entanglement, a quantitative measure of the discrepancy between hierarchical clusterings, at only 6% indicates a high degree of similarity. (For comparison, the entanglement value between the dendrogram for all expressed genes and that for transcript factors is 11% [*Schaum et al., 2018*]). As in the UMAP visualization, immune cells group together more closely in the splicing dendrogram. However, unlike the UMAP projection, we observe that several types of pancreatic cells cluster together with neurons, a cell type long believed to share an evolutionary origin (*Le Roith et al., 1982*). Notably, the left dendrogram in *Figure 7b* shows that hepatocytes are clear outliers in the expression latent space. We suspect this may be due to technical differences from using 96-well plates rather than the 384-well plates used for other cell types.

## B cell development in the marrow

We then focused on developing B cells from the bone marrow in *Tabula Muris*. In the splicing latent space, we found that immature B cells are harder to distinguish from the other B cell subpopulations (*Figure 8a*), reflecting less refined splicing programs or limitations in transcript capture efficiency. Immature B cells have also fewer differential splicing events when compared to the other stages of B cell development (*Figure 8b*). The top differential splicing events we identified throughout development displayed splicing trajectories mostly independent from the trajectories of gene expression (*Figure 8c*). We highlight alternative TSSs (one of them novel) in two transcription factors essential for B cell development: *Smarca4*, encoding BRG1 (*Bossen et al., 2015*; *Figure 8d*, *Figure 8—figure supplement 1*); and *Foxp1* (*Hu et al., 2006*; *Figure 8e*, *Figure 8—figure supplement 2*). While *Foxp1* expression peaks in pre-B cells and does not follow a monotonic trend over developmental stages, the alternative TSS is progressively included throughout B cell development. Combining gene-level expression with TSS usage, which can influence translation rate, provides a more nuanced

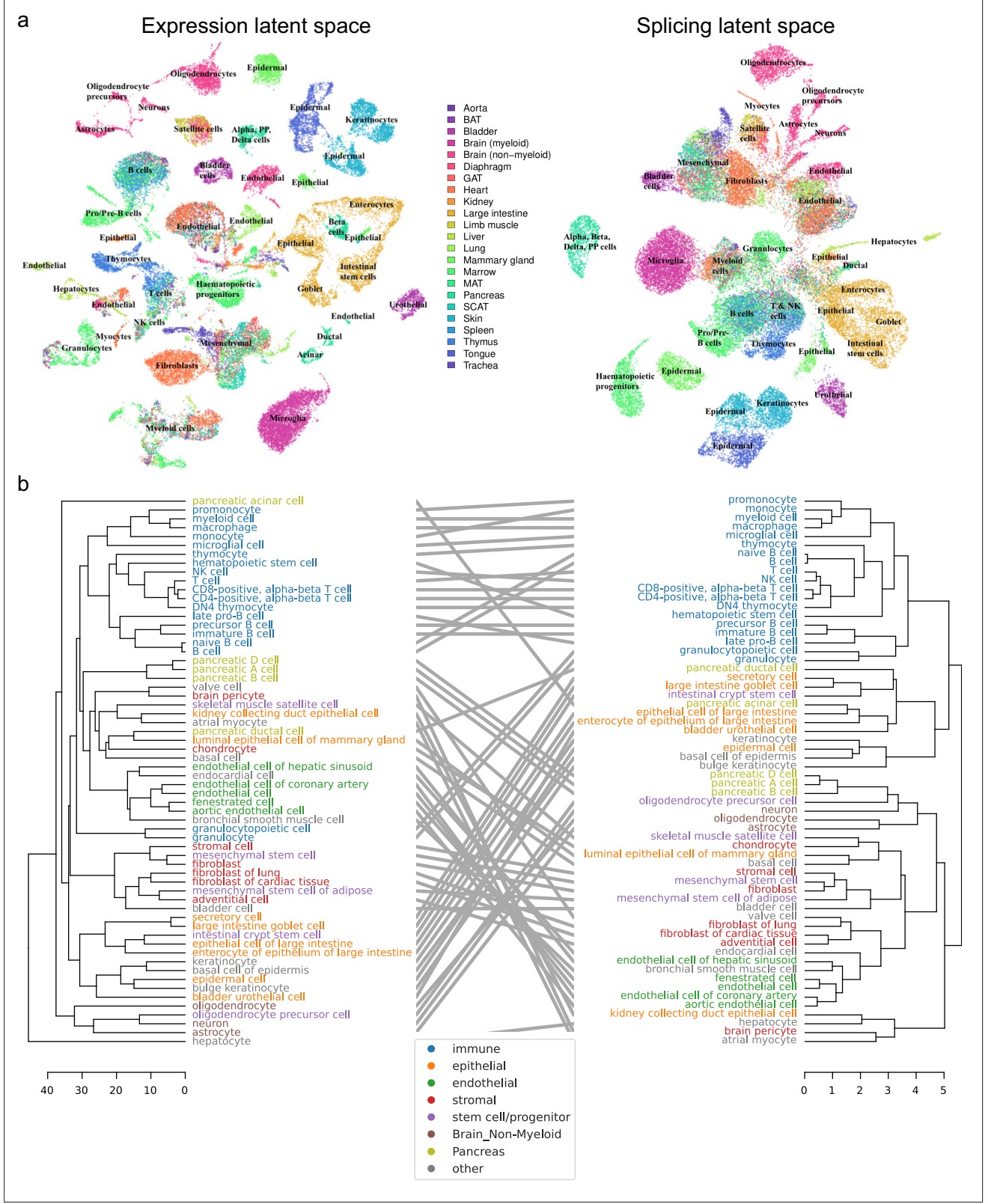

**Figure 7.** Global analysis of *Tabula Muris.* (**a**) UMAP visualization of the expression (left) and splicing (right) latent spaces. Each dot is a cell, colored by organ, and overlays indicate the primary cell type comprising that cluster. (**b**) Tanglegram comparing dendrograms of major cell types based on distances in the expression (left) and splicing (right) latent spaces, highlighting functional classes with specific colors.

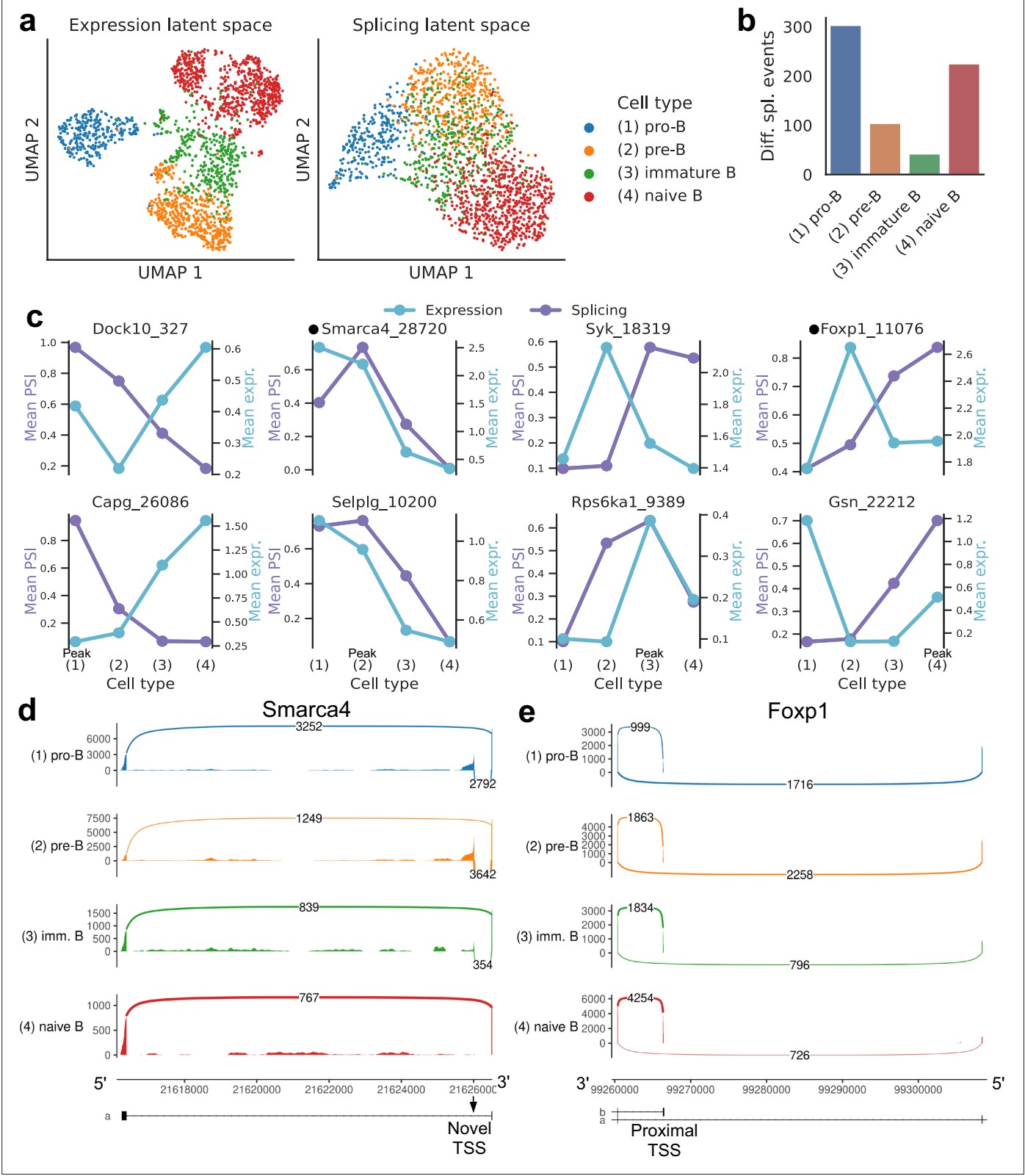

**Figure 8.** Splicing in developing marrow B cells from *Tabula Muris*. B cell developmental stages include pro-B, pre-B, immature B, and naive B. (**a**) Expression versus splicing latent space, as defined previously. In the splicing latent space, some cells types (pro-B) are better distinguished than others (immature B). (**b**) Number of differential splicing events when comparing a B cell stage vs. the rest. (**c**) PSI of some introns that are differentially spliced throughout development, together with expression of the respective genes (log-transformed normalized counts). Expression and splicing

*Figure 8 continued on next page*

*Figure 8 continued*

can have very different trajectories. (**d**) Sashimi plot of novel alternative transcription start site (TSS) in *Smarca4*. The novel TSS has maximum usage in pre-B cells, and then decays, while the expression peaks at pro-B cells. (**e**) Sashimi plot of an annotated alternative TSS in *Foxp1*. The proximal TSS in increasingly used as development progresses, while the expression peaks at pre-B cells.

The online version of this article includes the following source data and figure supplement(s) for figure 8:

**Source data 1.** Intron coordinates for panel (**c**).

**Figure supplement 1.** PSI distribution of Smarca4_28720.

**Figure supplement 2.** PSI distribution of Foxp1_11076.

characterization of the expression patterns of these important transcription factors. Some other differentially spliced genes with well-known roles in B cell development are *Syk* (***Cornall et al., 2000***), *Dock10* (***García-Serna et al., 2016***), *Selplg/Psgl-1* (***González-Tajuelo et al., 2020***), and *Rps6ka1* (***Stein et al., 2017***).

## Epithelial and endothelial cell types across organs

Having compared different cell types within organs, we analyzed putatively similar cell types which are present in multiple organs to investigate splicing variation associated with tissue environment and function. We find many alternative introns with strong PSI differences across epithelial cell types, including several which are novel (***Figure 9a***). Conversely, apart from those in the brain, endothelial cell types fail to display such striking differences (***Figure 9b***). These patterns are consistent with the UMAP projection and dendrogram, both of which suggested less heterogeneity among endothelial than epithelial cells (***Figure 7***).

Our analysis revealed a novel alternative TSS in *Itpr1* (***Figure 9c***, ***Figure 9—figure supplement 2***), an intracellular calcium channel in the endoplasmic reticulum, which regulates secretory activity in epithelial cells of the gastrointestinal tract (***Lemos et al., 2020***). This novel TSS yields a shorter protein isoform (full view in ***Figure 9—figure supplement 1***) which preserves the transmembrane domain, though it is unclear whether this isoform is functional. Notably, it is the predominant isoform in large intestine secretory cells, and these cells express *Itpr1* at the highest level among all epithelial cell types in the dataset. All nine novel alternative splicing events in ***Figure 9a*** are alternative TSSs, with four affecting the 5′ UTR and five affecting the coding sequence.

***Figure 9d*** (PSI distribution in ***Figure 9—figure supplement 3***) illustrates a complex alternative splicing event in *Khk* involving the well-studied exons 3a and 3c (***Hayward and Bonthron, 1998***). Khk catalyzes the conversion of fructose into fructose-1-phosphate, and the two protein isoforms corresponding to either exon 3a or 3c inclusion differ in their thermostability and substrate affinity (***Asipu et al., 2003***). While the literature describes these exons as mutually exclusive, the transcriptome reference includes transcripts where neither or both may be included. Although we did not find cell types with high inclusion rates for both exons, we did see multiple cell types where both exons are predominantly excluded, for example, epithelial cells from the large intestine. Other differentially spliced genes are involved in cellular junctions, which are particularly important in epithelial tissue. These include *Gsn*, *Eps8*, *Tln2*, *Fermt3*, and *Mapre2*.

## Comparison of selected tissues

Because of the breadth of the *Tabula Muris* dataset, we can look for general trends across a diverse array of tissues and cell types. ***Table 2*** summarizes differential expression and splicing for some of the cell types and tissues with the largest sample sizes. First, we note the intersection between the top 100 most differentially expressed and top 100 most differentially spliced genes (ranked by p-value) is consistently low. This means that most differentially spliced genes, which might be of critical importance in a biological system, will go unnoticed if a study only considers differential expression. Second, L5 IT neurons have a larger fraction of genes with differential splicing relative to the number of differentially expressed genes.

We found many more cell-type-specific differential splicing events in the cortex than in the marrow, as expected (***Yeo et al., 2004***), as well as a higher proportion of events involving novel junctions, which can reach 30% (***Figure 10a***). Differences in proportion of novel junctions should be interpreted with care, however, since they can be affected by sequencing depth and number of cells, both of

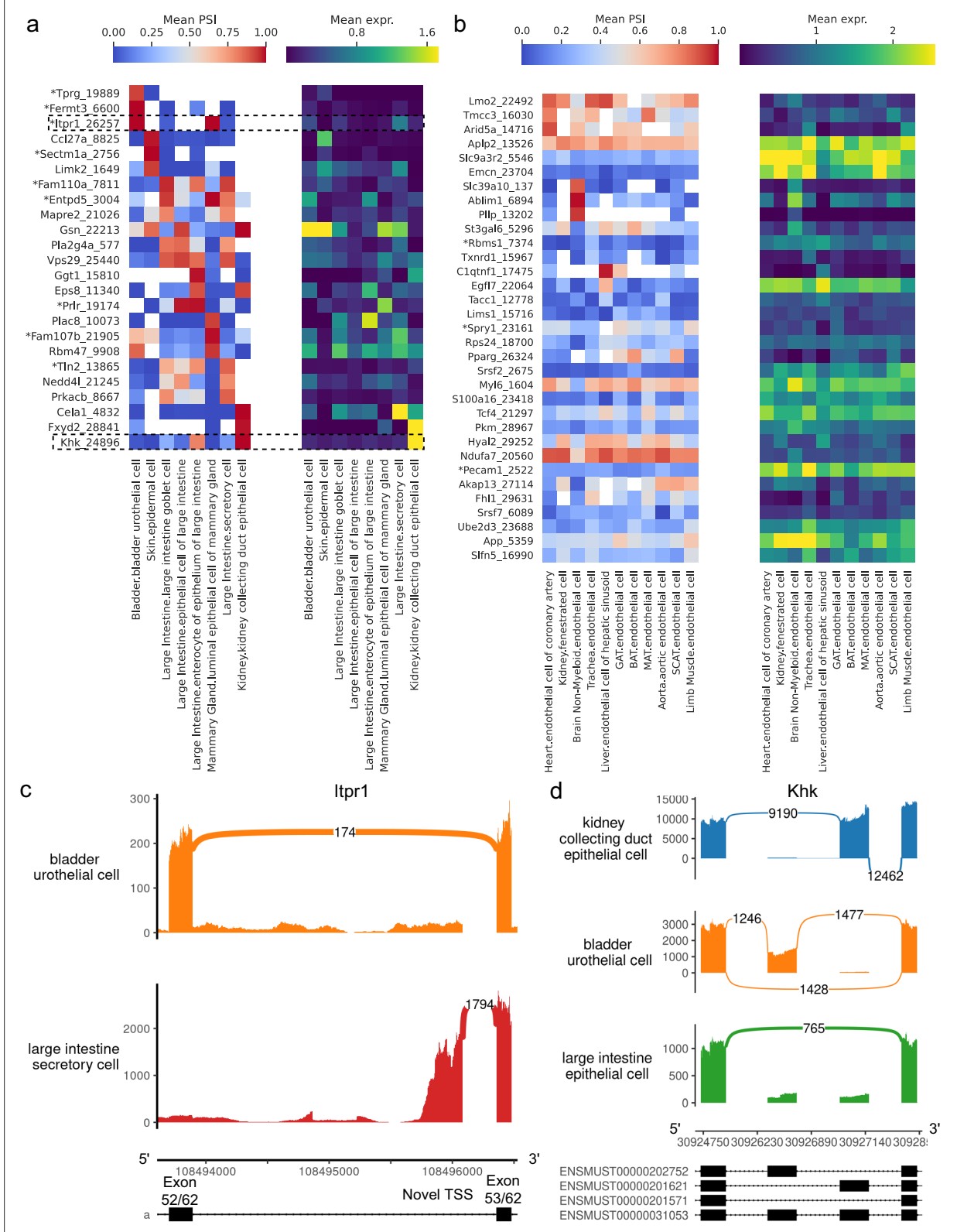

**Figure 9.** Alternative splicing patterns across epithelial and endothelial cell types. (**a–b**) PSI of selected introns (left) and expression (log-transformed normalized counts) of the corresponding genes (right) averaged across cell types. Novel intron groups are marked with (*). (**a**) Introns distinguishing epithelial cell types. (**b**) Introns distinguishing endothelial cell types. (**c**) Sashimi plot of an alternative TSS in *Itpr1*. (**d**) Sashimi plot of a complex alternative splicing event in *Khk*.

*Figure 9 continued on next page*

*Figure 9 continued*

The online version of this article includes the following source data and figure supplement(s) for figure 9:

**Source data 1.** Intron coordinates for panel (a).

**Source data 2.** Intron coordinates for panel (b).

**Figure supplement 1.** Full-gene view of novel alternative TSS in *Itpr1*.

**Figure supplement 2.** PSI distribution of Itpr1_26257.

**Figure supplement 3.** PSI distribution of Khk_24896.

which vary between the two tissues. Very similar patterns are seen when grouping differential splicing events that occur in the same gene (*Figure 10b*). Most differential splicing events that we detected with alternative introns fall in the coding portion of the gene, with high proportions in the 5' UTR (*Figure 10c*). This is a property of our quantification approach and does not reflect the total number of alternative splicing events in different gene regions; still, the relative proportion can be compared across tissues. We find an increased proportion of differentially spliced non-coding RNA in the cortex, the majority of which are previously unannotated events. To systematically evaluate how well cell types can be distinguished in the expression and splicing latent spaces, we calculated the ROC AUC score for the one-versus-all classification task for each cell type in each tissue using a binary logistic regression model (*Figure 10d*). Since cell type labels were defined using gene expression values, near-perfect classification is to be expected using the expression latent space. Classification based only on the splicing latent space is very good in general, suggesting that cell-type-specific differential splicing is rather pervasive. A few cell types were more challenging to classify correctly using splicing patterns alone. One such example is immature B cells, a reflection of the lower degree of separation observed in the embedding of *Figure 8a*.

## Finding splicing factors associated with specific alternative splicing events

Several splicing factors have been identified as regulators of specific alternative splicing events, but most regulatory interactions remain unknown (see *Vuong et al., 2016* for a review focused on the brain). To complement expensive and laborious knockout experiments, we sought to generate regulatory hypotheses by analyzing the correlation between splicing outcomes and splicing factor variation across cell types. Focusing on a subset of highly expressed genes in BICCN primary motor cortex neurons, we fit a sparse linear model regressing PSI of skipped exons on both expression and splicing

**Table 2.** Summary of differential expression and splicing for select cell types with the largest sample sizes.

The overlap between the top 100 differentially expressed genes and the top 100 differentially spliced genes is low, indicating that splicing provides complementary information. In addition, L5 IT neurons have a higher ratio of differentially spliced genes to differentially expressed genes than the other cell types. *Diff. spl. genes*: number of differentially spliced genes between the cell type and other cell types in the same tissue. *Diff. exp. genes*: number of differentially expressed genes between the cell type and other cell types in the same tissue. See Materials and methods for details on the tests for differential splicing and expression.

| Tissue | Total # cells | # cell types | Cell type | # cells | Diff. spl. genes | Diff. exp. genes | Ratio | Top-100 overlap |
|---|---|---|---|---|---|---|---|---|
| Brain Non-Myeloid | 3049 | 6 | Oligodendrocyte | 1390 | 880 | 8835 | 0.10 | 4 |
| Cortex | 6220 | 10 | L5 IT | 1571 | 1447 | 6402 | 0.23 | 2 |
| Heart | 4144 | 6 | Endothelial cell of coronary artery | 1126 | 465 | 7108 | 0.07 | 5 |
| Large Intestine | 3729 | 5 | Enterocyte of epithelium | 1112 | 586 | 10,786 | 0.05 | 2 |
| Marrow | 4783 | 10 | Hematopoietic stem cell | 1363 | 692 | 9909 | 0.07 | 2 |

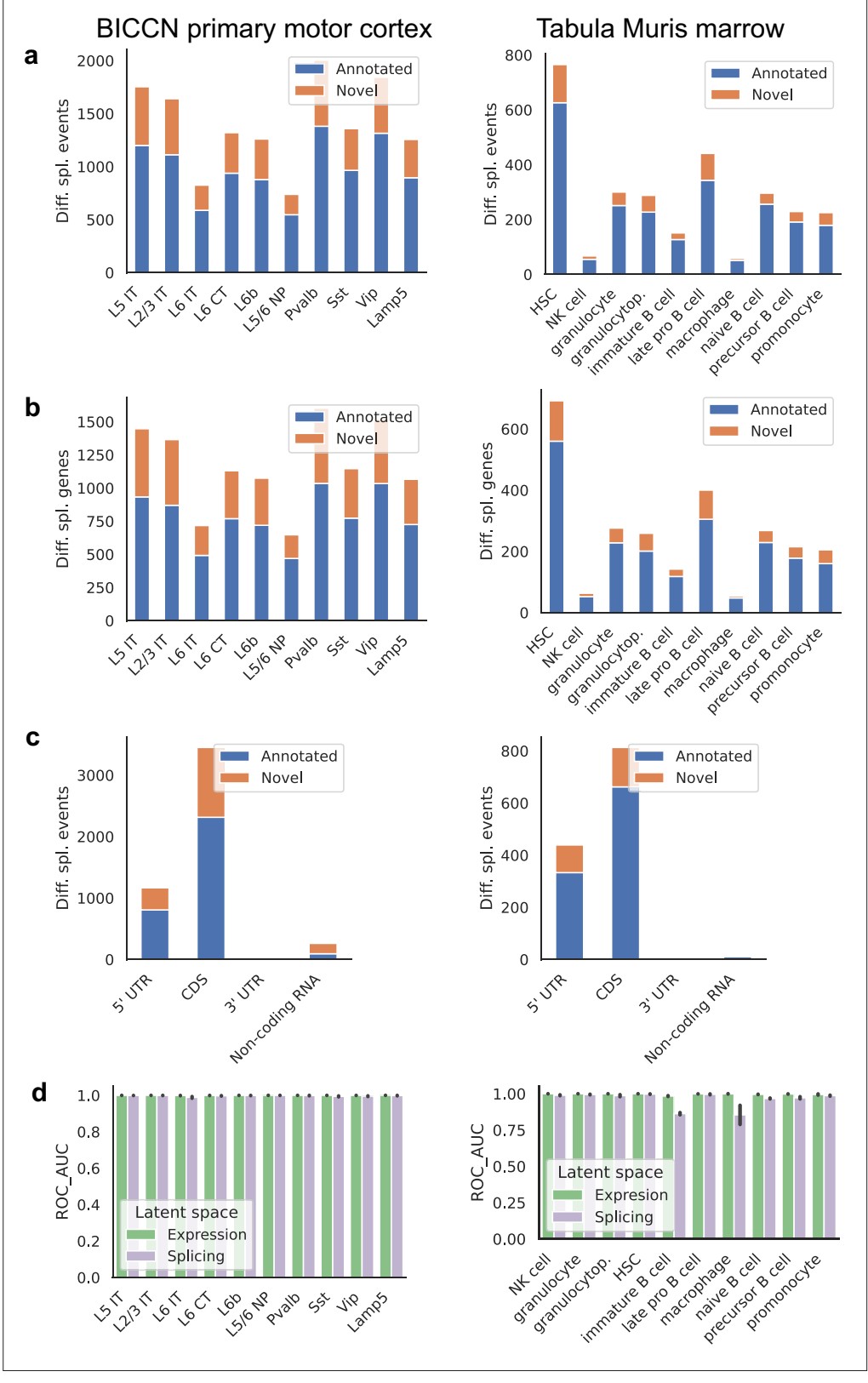

**Figure 10.** Patterns across tissues. (**a**) Number of differential splicing events detected in each cell type. Cortex cell types have more differential splicing events and larger proportions of novel events (those involving an intron absent from the reference). (**b**) Number of genes with a detected differential splicing event, for different cell types. (**c**) Number of differential splicing events in different gene regions aggregated over cell types (duplicate events

*Figure 10 continued on next page*

*Figure 10 continued*

removed). Cortex cell types have higher proportions of events in coding regions and non-coding RNAs. Note: y-axes are not on the same scale. (**d**) ROC AUC score for classification of each cell type versus the rest based on either the expression or splicing latent space, using logistic regression, training and testing in non-overlapping sets of individuals. The score for splicing-based classification is near-perfect in most cell types with some exceptions such as immature B cells in the marrow.

patterns of splicing factors (*Figure 11a* and *Figure 11—figure supplement 1*). Our model recovers several known regulatory interactions such as Khdrbs3/Slm2/T-Star's repression of splice site 4 (SS4) in neurexins, modulating their binding with post-synaptic partners (*Traunmüller et al., 2016*). Additionally, the proportion of a novel alternative TSS (though annotated in the human reference) in *Khdrbs3* (*Figure 11b*, *Figure 11—figure supplement 2*) is negatively associated with SS4 in *Nrxn1* and *Nrxn3*. This novel isoform lacks the first 30 amino acids of the Qua1 homodimerization domain and could affect dimerization, which modulates RNA affinity (*Feracci et al., 2016*). The model also recovers the known regulation of a skipped exon in *Camta1*, a transcription factor required for long-term memory (*Bas-Orth et al., 2016*), by Rbfox1 (*Pedrotti et al., 2015*). The skipping of exon 5 (E5) of *Grin1*, which controls long-term synaptic potentiation and learning (*Sengar et al., 2019*), is known to be regulated by Mbnl2 and Rbfox1 (*Vuong et al., 2016*). The model associates *Grin1* E5 PSI with the expression of *Rbfox1* but not *Mbnl2*; however, it does suggest an association with the PSI of two skipped exons in *Mbnl2* (*Figure 11c*, *Figure 11—figure supplements 3 and 4*) and further implicates the inclusion level of the novel alternative TSS in *Rbfox1* reported above (Rbfox1_26172, chr16:5763912–6173605, *Figure 6d*). These results help clarify the disparate impacts of expression and alternative splicing in splicing factors, and encourage the use of regression models to suggest candidate regulators of cell-type-specific alternative splicing. Such computationally generated hypotheses are particularly valuable for splicing events in splicing factors because of the heightened difficulty to experimentally perturb specific exons rather than whole genes.

## Discussion

In this study, we introduce scQuint, a toolkit for the quantification, visualization, and statistical inference of alternative splicing in full-length scRNA-seq data without the need for annotations. This allows us to successfully extend the analysis of two single-cell atlases to the level of alternative splicing, overcoming the usual technical challenges as well as coverage artifacts and incomplete annotations. Our results, which we make available for public exploration via interactive browsers, indicate the presence of strong cell-type-specific alternative splicing and previously unannotated splicing events across a broad array of cell types. In most cases, splicing variation is able to differentiate cell types just as well as expression levels. We also note a striking lack of overlap between the most strongly differentially expressed and spliced genes (*Table 2*), suggesting that expression and splicing are complementary rather than integrated processes. Moreover, this complementarity may also manifest temporally, as we show in developing B cells in the marrow. Another outstanding question is the functional significance of isoforms, and we find that most differential splice sites appear in the coding sequence with a sizeable minority also mapping to 5' UTRs. The apparent predilection for events to occur in these regions rather than 3' UTRs poses questions about the role of splicing in protein synthesis from translational regulation to the formation of polypeptide chains. Answering these questions requires a more precise understanding of how variation in UTRs and coding sequences affects final protein output as well as the biophysical characteristics of protein isoforms and their roles in different biological systems. These factors, combined with the large fraction of unannotated events in several cell types, should encourage tissue specialists to more deeply consider the contribution of transcript variation to cell identity and cell and tissue homeostasis.

Despite the clear association between splicing and cell identity, our analyses are yet to produce instances in which clustering in the splicing latent space reveals new cell subpopulations not visible in the expression latent space. This, of course, does not preclude the possibility in other settings where alternative splicing is known to be important, such as in specific developmental transitions or disease conditions. Nevertheless, our current experience leads us to believe that gene expression and splicing proportions provide two different projections of the same underlying cell state. Incidentally, RNA

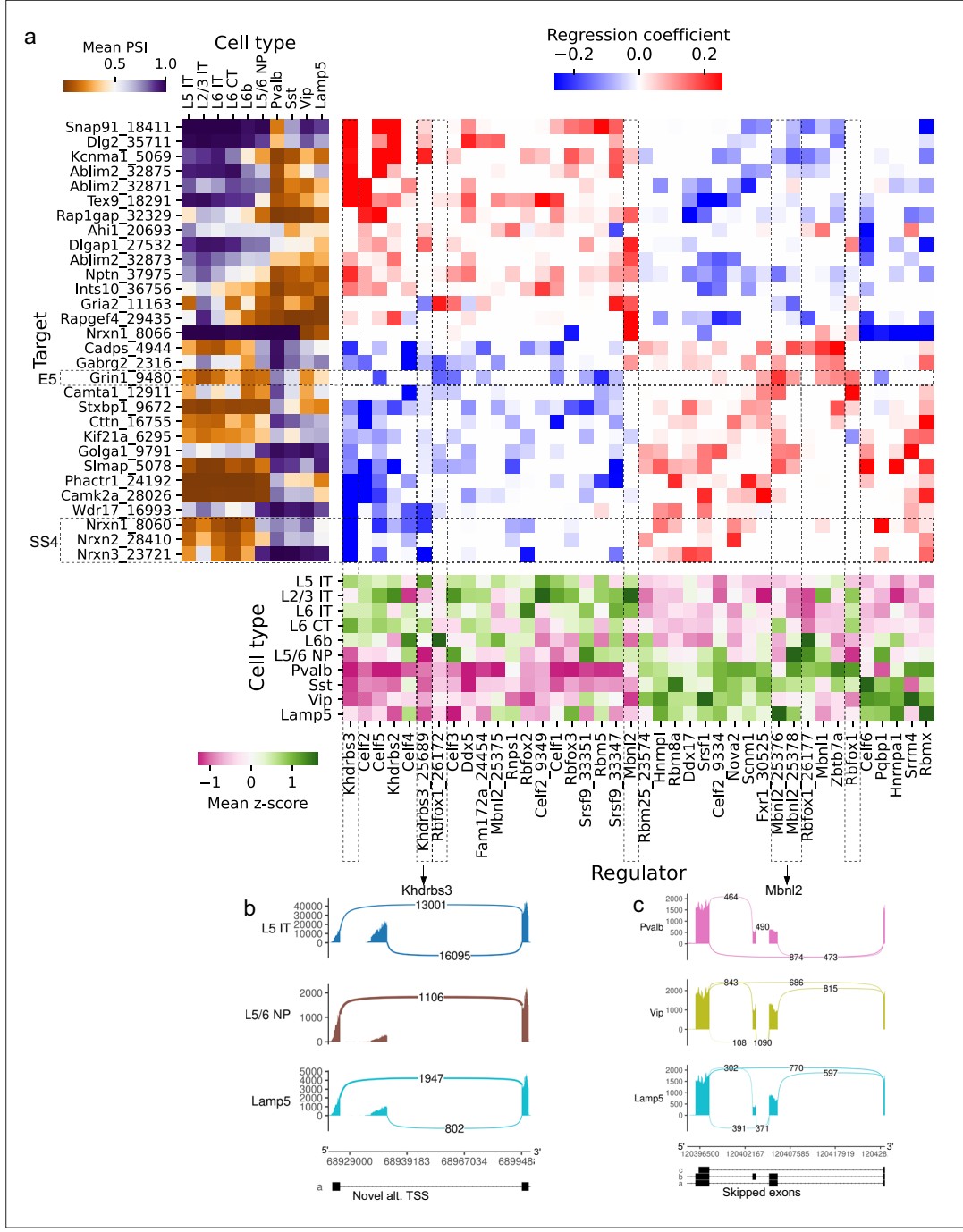

**Figure 11.** Associations between splicing factors and alternative splicing. (**a**) Regression analysis of exon skipping based on expression and splicing of splicing factors, using the BICCN mouse primary motor cortex dataset. Left panel: mean PSI of skipped exons across cell types. Bottom panel: mean z-scores of selected splicing factor features across cell types, including whole-gene expression (gene name) and PSI of alternative introns (gene name and numerical identifier). Center panel: regression coefficients (log-odds) of each splicing factor feature used to predict skipped exon PSI in our sparse Dirichlet-Multinomial linear model. (**b**) Novel alternative TSS in *Khdrbs3*. (**c**) Annotated skipped exons in *Mbnl2*.

The online version of this article includes the following source data and figure supplement(s) for figure 11:

**Source data 1.** Intron coordinates are available for panel (a).

**Figure supplement 1.** Full plot of associations between splicing factors and alternative splicing.

**Figure supplement 2.** PSI distribution of Khdrbs3_25689.

*Figure 11 continued on next page*

*Figure 11 continued*

**Figure supplement 3.** PSI distribution of Mbnl2_25376.

**Figure supplement 4.** PSI distribution of Mbnl2_25378.

Velocity (*La Manno et al., 2018*) estimates can be distorted by alternative splicing, and (*Bergen et al., 2020*) discuss incorporating isoform proportions into the model as a future direction.

To support our understanding of cell-type-specific splicing, we implemented a regularized generalized linear regression model which exploits the natural variation of splicing factors in different cell types. We recovered a number of previously identified (via knockout experiments) regulatory interactions and propose novel regulatory interactions involving genes known to play important regulatory roles. A key component of our analysis is the decision to include both the expression and alternative splicing patterns of splicing factors as features in the model. Consequently, we infer that several alternative splicing events in splicing factors themselves (some previously unannotated) contribute to their regulatory activity. Our model thus provides several opportunities for follow-up and does so with an increased granularity that distinguishes between effects due to expression and splicing differences. To facilitate further exploration of these data, we have uploaded our results to cell and genome browsers (linked at https://github.com/songlab-cal/scquint-analysis, (*Benegas, 2021a* copy archived at swh:1:rev:97dc31babf2a585666af4a38b1e4aa59a92bbf87)).

Our experience analyzing these large datasets, initially with prior methods and then scQuint, has led to a series of general observations regarding the analysis of splicing in scRNA-seq data. As most analyses use full-length short-read protocols because of the cost of long-read data and the necessary focus on the 3' end of transcripts in most UMI-based techniques, we restrict our attention to the full-length short-read setting and its incumbent challenges. For example, low transcript capture efficiency introduces additional technical noise into isoform quantification (*Arzalluz-Luque and Conesa, 2018*; *Westoby et al., 2020*; *Buen Abad Najar et al., 2020*), and incomplete transcriptome annotations result in discarded reads and reduced sensitivity to cross-cell differences (*Westoby et al., 2020*). Nonetheless, we considered several methods (summarized in *Appendix 1—table 1*) to analyze transcript variation in short-read, full-length scRNA-seq. We found each of the classes of current methods to be problematic in the context of our datasets for varying reasons. Methods which depend on transcript annotations (*Bray et al., 2016*; *Qiu et al., 2017*; *Huang and Sanguinetti, 2017*; *Hu et al., 2020*; *Yan et al., 2015*; *Wen et al., 2020*; *Liu et al., 2021*; *Huang and Sanguinetti, 2021*; *Tekath and Dugas, 2021*) cannot easily identify unannotated alternative splicing events. In large collections of previously unsurveyed cell types, these may comprise a sizable fraction of events. Indeed, we found up to 30% of differential splicing events were unannotated in certain cell types. Annotation-free approaches are also available, but they either do not provide a formal statistical test for differential transcript usage across conditions (*Song et al., 2017*; *Ling et al., 2020*; *Nip et al., 2020*; *Welch et al., 2016*), or only do so in a specialized manner (*Matsumoto et al., 2020*), reducing their potential impacts. Finally, methods' different approaches to quantification are affected by coverage biases to varying degrees. Some methods may thus lead to erroneous inference of cell clusters due to technical rather than biological variation. Until the prevalence and severity of coverage biases are better understood, we advocate quantifying transcript variation in a robust manner.

Recent and future experimental advances will catalyze the study of isoform variation in single cells. For instance, Smart-seq3 (*Hagemann-Jensen et al., 2020*) allows sequencing of short reads from the entire length of a gene together with unique molecular identifiers, improving mRNA capture and allowing for the filtering of PCR duplicates; however, experiments show that less than 40% of reads can be unambiguously assigned to a single (annotated) isoform. Ultimately, long-read scRNA-seq will provide the definitive picture of isoform variation between cells. Until then, there is much biology to be studied using short-read protocols, and variation at the transcript level should not be disregarded.

## Materials and methods
### Datasets
*Tabula Muris* data (*Schaum et al., 2018*) have accession code GSE109774. Cells were filtered to those from 3-month-old mice present in this collection: https://czb-tabula-muris-senis.s3-us-west-2.

BICCN Cortex data (*Yao et al., 2021*) were downloaded
from https://assets.nemoarchive.org/dat-ch1nqb7 and filtered as in *Booeshaghi et al., 2021*.

## Simulation

A preliminary set of exon skipping events was obtained by running briekit-event from the BRIE2 software package. For each event, one pair of transcripts was selected if they only differed on the skipped exon, resulting in 561 pairs, each from a different gene. Reads were simulated using Polyester (*Frazee et al., 2015*), which allows us to control overdispersion and induce different kinds of biases. For roughly half of the genes, differential transcript usage (DTU) was induced by overexpressing one transcript 1.5-fold in one of the two conditions. The number of reads was generated using a highly overdispersed negative binomial distribution with variance equal to eight times the mean. To simulate coverage decay in one of the conditions, the option bias="cdnaf" was added. To ensure coverage decays as a function of absolute distance to the 3' end of the transcript, reads were generated no farther away from the 3' than the minimum of the lengths of the two alternative transcripts. The Area Under the Receiver Operating Characteristic Curve (ROC AUC) for classifying genes into DTU vs. non-DTU was computed using the p-values from each method, excluding genes that were not tested by a given method (e.g., because of a minimum reads threshold).

## Quantification

The bioinformatic pipeline was implemented using Snakemake (*Köster and Rahmann, 2012*). Raw reads were trimmed from Smart-seq2 adapters using Cutadapt (*Martin, 2011*) before mapping to the GRCm38/mm10 genome reference (https://hgdownload.soe.ucsc.edu/goldenPath/mm10/chromosomes/) and the transcriptome reference from Ensembl release 101 (ftp://ftp.ensembl.org/pub/release-101/gtf/mus_musculus/Mus_musculus.GRCm38.101.gtf.gz). Alignment was done using STAR (*Dobin et al., 2013*) in two-pass mode allowing novel junctions as long as they were supported by reads with at least 20 base pair overhang (30 if they are non-canonical) in at least 30 cells. Also, multi-mapping and duplicate reads were discarded using the flag --bamRemoveDuplicatesType Unique Identical (while this can remove duplicates from the second PCR step of Smart-seq, it will not remove duplicates from the first PCR step). Soft-clipped reads were removed as well. Additionally, reads were discarded if they belonged to the ENCODE region blacklist (*Amemiya et al., 2019*) (downloaded from https://github.com/Boyle-Lab/Blacklist/raw/master/lists/mm10-blacklist.v2.bed.gz).

Gene expression was quantified using featureCounts (*Liao et al., 2014*), and total-count normalized such that each cell had 10,000 reads (as in the Scanpy (*Wolf et al., 2018*) tutorial). Intron usage was quantified using split reads with an overhang of at least six base pairs. Introns were discarded if observed in fewer than 30 cells in *BICCN Cortex* or 100 cells in *Tabula Muris*. Introns were grouped into alternative intron groups based on shared 3' splice acceptor sites. Introns not belonging to any alternative intron group were discarded. Additionally, we decided to subset our analysis to introns with at least one of their donor or acceptor sites annotated, so we could assign a gene to each intron and facilitate interpretation for our specific analyses.

## Dimensionality reduction

To run PCA, we worked with alternative intron proportions (PSI, Percent Spliced In) rather than their absolute counts, as the latter would be confounded by gene expression differences. We first introduce some notation:

- $c$: cell identifier
- $g$: intron group identifier
- $\vec{y}_g^{(c)}$: vector of counts of introns in intron group $g$ and cell $c$
- $\text{normalize}(\vec{x}) = \frac{\vec{x}}{sum(\vec{x})}$: function to divide each entry of a vector by the total sum.

Then, PSI can be defined as:

$$\overrightarrow{\text{PSI}}_g^{(c)} = \text{normalize}\left(\vec{y}_g^{(c)}\right)$$

**Table 3.** VAE hyperparameters.

| Dataset | Decoder | Layers | $\sigma$ | Latent dimension |
|---|---|---|---|---|
| BICCN Cortex | Linear | 1 | 26.8 | 18 |
| Tabula Muris | Non-linear | 2 | - | 34 |

However, given the sparsity of single-cell data, a very high proportion of alternative intron groups will have no reads in a given cell, leaving PSI undefined. More generally, an intron group may contain few reads, resulting in defined but noisy PSI estimates. To navigate this issue, we introduce a form of empirical shrinkage towards a central value. We first define the 'global PSI' by aggregating reads from all cells and normalizing. Then, we add this global PSI as a pseudocount vector to each cell before re-normalizing to obtain each cell's shrunken PSI profile (these are non-uniform pseudocounts adding up to one).

$$\overrightarrow{\text{PSI}}_g^{(\text{global})} = \text{normalize}\left(\sum_c \vec{y}_g^{(c)}\right)$$

$$\overrightarrow{\text{SMOOTHED\_PSI}}_g^{(c)} = \text{normalize}\left(\vec{y}_g^{(c)} + \overrightarrow{\text{PSI}}_g^{(\text{global})}\right)$$

We then run standard PCA on the cell-by-intron-smoothed PSI matrix.

The VAE was implemented using PyTorch (*Paszke et al., 2019*) and scvi-tools (*Gayoso et al., 2021*). The following is the generative model, repeated for each cell (we drop the superscript indexing the cell in $\vec{z}$, $\vec{p}$, $\vec{y}$ and $\vec{n}$):

1. Sample the latent cell state $\vec{z} \sim \text{Normal}(0, \text{I})$
2. For each intron group $g$:
   a. Obtain the underlying intron proportions: $\vec{p}_g = \text{softmax}(f_g(\vec{z}))$
   b. Sample the intron counts conditioning on the total observed $n$g: $\vec{y}_g | n_g \sim \text{DirichletMultinomial}\left(n_g, \alpha_g \cdot \vec{p}_g\right)$

Here $f_g$, known as the decoder, can be any differentiable function, including linear mappings and neural networks. $\alpha_g$ is a scalar controlling the amount of dispersion. We optimize a variational posterior on cell latent variables $q(z|y)$ (Gaussian with diagonal covariance, given by an encoder neural network) as well as point estimates of global parameters $f_g$, $\alpha_g$. The encoder takes as input the smoothed PSI values, as in PCA, but the likelihood is based on the raw intron counts. The objective to maximize is the evidence lower bound (ELBO), consisting of a reconstruction term and a regularization term:

$$\text{ELBO}(y) = \mathbb{E}_{z \sim q(z|y)}[\log p(y|z)] - \text{KL}(q(z|y)\|p(z)),$$

where $\text{KL}(\cdot\|\cdot)$ denotes the Kullback–Leibler divergence. Optimization is performed using Adam (*Kingma and Ba, 2015*), a stochastic gradient descent method. To avoid overfitting in cases of relatively few cells with respect to the number of features, we considered a linear decoder (*Svensson et al., 2020*), as well as a $\text{Normal}(0, \sigma)$ prior on the entries of the decoder matrix. Hyperparameters were tuned using reconstruction error on held-out data and are described in *Table 3*.

## Differential splicing test

Our differential splicing test across conditions (such as cell types) is based on a modified version of the Dirichlet-Multinomial Generalized Linear Model proposed in LeafCutter (*Li et al., 2018*) for bulk RNA-seq. For each intron group $g$ with $L$ alternative introns:

- $\vec{y}_g$ is a vector of counts for each of the $L$ introns;
- The independent variable, $x$, equals 0 in one condition and 1 in the other;
- $\vec{a}_g, \vec{b}_g \in \mathbb{R}^{L-1}$ are the intercept and coefficients of the linear model;
- $\alpha_g \in \mathbb{R}$ is a dispersion parameter shared across conditions; and
- the function softmax : $(z_1, \ldots, z_{L-1}) \mapsto \left(\frac{e^{z_1}}{1+\sum_{i=1}^{L-1} e^{z_i}}, \ldots, \frac{e^{z_{L-1}}}{1+\sum_{i=1}^{L-1} e^{z_i}}, \frac{1}{1+\sum_{i=1}^{L-1} e^{z_i}}\right)$ maps from $\mathbb{R}^{L-1}$ to the $(L-1)$-dimensional probability simplex.

The Dirichlet-Multinomial Generalized Linear Model then proceeds as follows:

1. Obtain the underlying intron proportions: $\vec{p}_g = \text{softmax}(\vec{a}_g + \vec{b}_g x)$
2. Sample the intron counts conditioned on the total observed, $n$g: $\vec{y}_g | n_g \sim \text{DirichletMultinomial}\left(n_g, \alpha_g \vec{p}_g\right)$

We implemented this model in PyTorch and optimized it using L-BFGS (*Liu and Nocedal, 1989*). To test for differential splicing across the two conditions, we compare the following two hypotheses:

Null hypothesis $H_0$:$\vec{b}_g = \vec{0}$
Alternative hypothesis $H_1$:$\vec{b}_g \neq \vec{0}$

We use the likelihood-ratio test, the test statistic for which is asymptotically distributed as a $\chi^2$ random variable with $L-1$ degrees of freedom under $H_0$. Finally, we correct p-values for multiple testing using the Benjamini-Hochberg FDR procedure (*Benjamini and Hochberg, 1995*).

The differences with LeafCutter are the following:

- LeafCutter groups introns that share a 5' donor or 3' acceptor site while scQuint groups introns that share a 3' acceptor site.
- LeafCutter has a vector of concentration parameters, one for each intron, while scQuint uses a single concentration parameter per intron group.
- The LeafCutter and scQuint optimization procedures were implemented separately and differ in initialization strategies as well as L-BFGS hyperparameters.

## Latent space analysis

The expression latent space was obtained by running PCA with 40 components on log-transformed and normalized gene expression values. The splicing latent space was obtained by running the VAE on the alternative intron count matrix (or equivalent features, e.g., Kallisto transcript counts, DEXSeq exon counts). Both latent spaces were visualized using UMAP (*McInnes et al., 2018*). In the comparison of *Figure 1*, we used our own implementation of the quantifications proposed by ODEGR-NMF, DEXSeq, and DESJ for ease of application to large single-cell datasets.

Dendrograms were constructed using hierarchical clustering (R function hclust) based on euclidean distance between the median latent space embedding of cells of each type. Tanglegram and entanglement were calculated using the dendextend R package, with the step2side method, as also described in *Schaum et al., 2018*.

Reported scores for cell type classification within a tissue were obtained by running a binary logistic regression classifier over different splits of cells into train and test sets. To assess generalization across individuals, we ensured the same individual was not present in both train and test sets.

## Cell-type-specific differential splicing

For differential splicing testing between a given cell type and the rest of the tissue, we only considered introns expressed in at least 50 cells and intron groups with at least 50 cells from both of the conditions. We called an intron group 'differentially spliced' if it was both statistically significant using a 5% FDR and if it contained an intron with a PSI change greater than 0.05. We considered a differentially spliced intron group as unannotated if it contained an unannotated intron with a PSI change greater than 0.05. Differential expression was performed using the Mann-Whitney test. A gene was considered differentially expressed if it was statistically significant using a 5% FDR and if the fold change was at least 1.5.

For selection of marker genes or introns, we proceeded in a semi-automated fashion. For each cell type, we first filtered to keep only significant genes or introns and then ranked them by effect size. We picked a certain number of genes or introns from the top of this list for each cell type, while ensuring there were no repetitions.

## Splicing factor regression analysis

We obtained 75 mouse splicing factors using the Gene Ontology term 'alternative mRNA splicing, via spliceosome' (http://amigo.geneontology.org/amigo/term/GO:0000380). A skipped exon annotation, processed by BRIE (*Huang and Sanguinetti, 2017*), was downloaded from https://sourceforge.net/projects/brie-rna/files/annotation/mouse/gencode.vM12/SE.most.gff3/download. Instead of using single cells as replicates, we partitioned the BICCN primary motor cortex dataset into roughly

200 clusters of 30 cells each that were pooled to create pseudobulks, aiming to reduce variance in the expression and splicing of splicing factors used as covariates in the model. We filtered target exon skipping events to those defined in at least 95% of the replicates, and those having a PSI standard deviation of at least 0.2. We used log-transformed normalized expression and PSI of alternative splicing events as input features. We chose to keep the PSI of only one intron per intron group to avoid the presence of highly correlated features and improve clarity, even if some information from non-binary events is lost. Input features were filtered to those having standard deviation of at least 0.05, and then standardized. A lasso Dirichlet-Multinomial GLM was fit to the data (in this instance, the model reduces to a Beta-Binomial because skipped exons are binary events), with the sparsity penalty selected via cross-validation. As a first approach, we fit a regular lasso linear regression model on PSI instead of raw counts, resulting in roughly similar patterns in the coefficients. *Figure 11c* shows the coefficients of the lasso Dirichlet-Multinomial model for the top 30 targets with the highest variance explained by the regular lasso model, all above 68%.

## Code and data availability

scQuint is implemented in Python and is available at https://github.com/songlab-cal/scquint, (*Benegas, 2021b* copy archived at swh:1:rev:a9db6454e13d42af25f47deee19e201e74d2bdd0). Differential splicing results and access to cell and genome browsers, together with the code to reproduce our results, are available at https://github.com/songlab-cal/scquint-analysis, (*Benegas, 2021c* copy archived at swh:1:rev:97dc31babf2a585666af4a38b1e4aa59a92bbf87). Processed alternative intron count matrices are provided in the AnnData format (anndata.readthedocs.ioanndata.readthedocs.io) for easy manipulation with Scanpy (*Wolf et al., 2018*), Seurat (*Stuart et al., 2019*), and other tools.

## Acknowledgements

We would like to thank Angela Oliveira Pisco, Spyros Darmanis, and Kif Liakath-Ali for helpful discussions. We also thank the Chan Zuckerberg Biohub for hosting our cell×gene sessions and Aaron McGeever for assistance. This research is supported in part by grant number R35-GM134922 from NIH and grant number CZF2019-002449 from the Chan Zuckerberg Initiative Foundation. YSS is a Chan Zuckerberg Biohub Investigator.

## Additional information

### Funding

| Funder | Grant reference number | Author |
|---|---|---|
| National Institutes of Health | R35-GM134922 | Gonzalo Benegas<br>Yun S Song |
| Chan Zuckerberg Initiative | CZF2019-002449 | Gonzalo Benegas<br>Yun S Song |

The funders had no role in study design, data collection and interpretation, or the decision to submit the work for publication.

### Author contributions

Gonzalo Benegas, Conceptualization, Investigation, Methodology, Software, Visualization, Writing - original draft; Jonathan Fischer, Conceptualization, Investigation, Supervision, Writing - original draft; Yun S Song, Conceptualization, Funding acquisition, Investigation, Supervision, Writing – review and editing

### Author ORCIDs

Gonzalo Benegas http://orcid.org/0000-0002-6639-4394
Jonathan Fischer http://orcid.org/0000-0002-1165-9930
Yun S Song http://orcid.org/0000-0002-0734-9868

**Decision letter and Author response**
Decision letter https://doi.org/10.7554/eLife.73520.sa1
Author response https://doi.org/10.7554/eLife.73520.sa2

# Additional files

## Supplementary files
• Transparent reporting form

## Data availability

All data analyzed in this study are publicly available and URL links are provided in the Materials and methods section of our manuscript. Our source code as well as all results represented in figures and tables are publicly available on our lab's GitHub repositories: https://github.com/songlab-cal/scquint, (copy archived at swh:1:rev:a9db6454e13d42af25f-47deee19e201e74d2bdd0) and https://github.com/songlab-cal/scquint-analysis, (copy archived at swh:1:rev:97dc31babf2a585666af4a38b1e4aa59a92bbf87).

The following previously published datasets were used:

| Author(s) | Year | Dataset title | Dataset URL | Database and Identifier |
|---|---|---|---|---|
| Schaum et al | 2018 | Tabula Muris | https://www.ncbi.nlm.nih.gov/geo/query/acc.cgi?acc=GSE109774 | NCBI Gene Expression Omnibus, GSE109774 |
| Yao et al | 2021 | BRAIN Initiative Cell Census Network Cortex | https://assets.nemoarchive.org/dat-ch1nqb7 | nemoarchive, dat-ch1nqb7 |

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

# Appendix 1

## Overview of available methods for alternative splicing analysis in full-length scRNA seq data

Due to experimental considerations, the analysis of transcript variation in 10x Chromium data is mostly restricted to the 3' end of genes; in contrast, Smart-seq2 and other full-length, short-read protocols theoretically enable characterization of transcript variation along the whole gene. Nevertheless, numerous challenges impede such analyses in practice. For example, low transcript capture efficiency introduces additional technical noise into transcript quantification (*Arzalluz-Luque and Conesa, 2018*; *Westoby et al., 2020*; *Buen Abad Najar et al., 2020*), and incomplete transcriptome annotations result in discarded reads and reduced sensitivity to cross-cell differences (*Westoby et al., 2020*). Some authors have even recommended avoiding the analysis of alternative splicing in single-cell RNA sequencing (scRNA-seq) data until such obstacles can be suitably overcome (*Westoby et al., 2020*). Despite these difficulties, several methods (summarized in *Appendix 1—table 1*) have sought to analyze transcript variation in short-read, full-length scRNA-seq. Many methods, including kallisto (*Bray et al., 2016*), Census (*Qiu et al., 2017*), BRIE (*Huang and Sanguinetti, 2017*), SCATS (*Hu et al., 2020*), Quantas (*Yan et al., 2015*), VALERIE (meant only for visualization) (*Wen et al., 2020*), DESJ (*Liu et al., 2021*), BRIE2 (*Huang and Sanguinetti, 2021*) and DTUrtle (*Tekath and Dugas, 2021*), depend on transcript annotations and consequently cannot easily identify unannotated alternative splicing events, which may comprise a sizable fraction of events. Currently available annotation-free methods, such as ODEGR-NMF (*Matsumoto et al., 2020*), Expedition (*Song et al., 2017*), ASCOT (*Ling et al., 2020*), SingleSplice (*Welch et al., 2016*) and RNA-Bloom (*Nip et al., 2020*), do not provide a statistical test for differential transcript usage across conditions. *Appendix 1—table 1* summarizes this information and makes the comparison of different methods easier.

**Appendix 1—table 1.** Summary of methods available to analyze transcript variation in short-read full-length scRNA-seq.

*Annotation-free*: Does quantification require an accurate transcriptome reference? *Differential transcript usage*: Does the method provide a two-sample test for differences in transcript proportions? Some methods, denoted by (*), provide other statistical tests. Quantas requires cells to be aggregated into known subgroups of each group and therefore does not perform a test at the single-cell level. SingleSplice tests for alternative splicing within a single population. kallisto and ODEGR-NMF test for differential transcript expression, i.e., changes in absolute transcript expression rather than their proportions. Census tests for differential transcript usage along a pseudotime trajectory.

| Method | Annotation-free | Differential transcript usage |
|---|---|---|
| Quantas [80] | | * |
| SingleSplice [76] | ✓ | * |
| kallisto [10] | | * |
| Census [56] | | * |
| BRIE [27] | | ✓ |
| Expedition [60] | ✓ | |
| ODEGR-NMF [46] | ✓ | * |
| SCATS [26] | | ✓ |
| RNA-Bloom [49] | ✓ | |
| ASCOT [41] | ✓ | |
| DESJ [43] | | ✓ |
| BRIE2 [28] | | ✓ |
| DTUrtle [65] | | ✓ |
| scQuint | ✓ | ✓ |

