## [Editor Report]

This paper presents a new method to study known and novel alternative splicing events at the single-cell level and perform differential analysis across cell types. The method addresses current challenges in the analysis of splicing in single cells related to technical variation and experimental biases. Performing one of the most comprehensive studies to date with data from different mice, this work expands the body of splicing events that potentially define individual cell types.

---

## [Decision Letter]

**Decision letter after peer review:**

Thank you for submitting your article "Robust and annotation-free analysis of alternative splicing across diverse cell types in mice" for consideration by *eLife*. Your article has been reviewed by 2 peer reviewers, and the evaluation has been overseen by a Reviewing Editor and Senior Editor. The reviewers have opted to remain anonymous.

The reviewers have discussed their reviews with one another, and it was agreed that a resubmission that fully addresses all of the concerns raised would be suitable for further consideration for publication in *eLife*. The Reviewing Editor has drafted the following to help you prepare a revised submission.

Essential revisions:

The reviewers have indicated that although the work might be of interest to researchers working in alternative splicing, the method requires significant additional testing and benchmarking, and the novelty of the findings must be made more clear. The reviewers have provided multiple suggestions to improve this and other aspects of the manuscript.

*Reviewer #1 (Recommendations for the authors):*

Authors have applied their method to two big scRNA-Seq datasets and have reported multiple biological discoveries from their computational analysis. However, the presentation and validation of the results should be improved. I elaborate on my comments below:

One of my major concerns is about the evaluation and benchmarking analysis of scQuint. Authors have particularly reviewed some of the existing methods in the appendix, but they provided no comparison between the performance of scQuint and those methods. Particularly authors have mentioned on page 18 that they have previously analyzed these datasets with prior methods, but they did not provide any comparison between their findings and those by other methods. In its current form, it is extremely difficult to judge the sensitivity and specificity of scQuint or whether it is a new contribution to the field. Thus, the paper's contribution is to run a standard, published analysis on a single cell dataset. There is no functional or experimental validation to support or refute the findings. Further, there is no computational validation, in terms of testing whether the predictions in this dataset hold in other data.

One limitation of the current method is more statistical tests which could lower the statistical power due to multiple hypothesis testing issue, as it needs to perform a separate test for each pair of gene/cell type compared to a test for each gene that some other methods need for finding "genes" with cell-type-specific splicing.

Authors have used the same model as in leafcutter for their analysis. However, they claim that they are getting better p-value and clustering results compared to leafcutter. It is not clear why their method should perform better than leafcutter.

One of the major advantages of the tabula muris dataset is that it contains data from multiple mouse individuals (i.e., biological replicates), which can be leveraged to show the reproducibility of the biological findings across biological replicates. However, authors did not take advantage of this in presenting their results. I highly recommend that authors show that their results can be replicated across mouse individuals, by visualizing their results as stratified by donor ID. Reproducibility is important for distinguishing between a real reproducible biological signal and a biological/technical noise particularly for the unannotated splicing events as they might be a product of splicing noise.

Authors have applied their method to only SS2 and not to any 10x data. I believe that the tabula muris dataset contains 10x data as well. While I agree that 10x is more challenging than SS2 for splicing analysis, it is still a valuable resource for splicing analysis as it has higher throughput compared to SS2 and can better capture rare cell types. I recommend that authors comment on the applicability of their method to 10x in the paper and, if their method is applicable, show how their current results compare to the results based on 10x data.

Since the paper is about analyzing splicing in single cells, I think it is extremely valuable to show the variation at the "single-cell level" (rather than pseudobulked cell-type level) via box or violin plots. This is extremely important as it is not clear from the current plots (e.g., figure 5 c,d,e or figure 8c,d) that the splicing event was observed in how many cells in each cell type and what is the range of read counts per single cell in each cell type. As I mentioned earlier it is extremely difficult to judge the reproducibility and single-cell variation of the visualized splicing events in figures as the data is aggregated across all cells within the cell type from separate donors.

Authors mention that there is little overlap between differentially expressed and differentially spliced genes but on the other hand they say that the clusters based on splicing and expression latent space are highly consistent with each other. I think they should comment on why this is possible, is it because the same cluster has different markers in each space. If so, is it possible to highlight a few clusters and show their marker genes based on splicing and expression changes?

On page 9, authors say that they detected thousands of cell-type-specific events; however, they do not provide more specifics about these events? How many events exactly? Across how many distinct genes (also what fraction of genes, and is this fraction with previous studies?)? And distinct cell types? Also, it is not clear how the examples in figure 5 were chosen? Are they among the top genes? What are the top genes? Are they genes known to have cell-type-specific splicing?

The paper lacks any experimental validation on the discovered splicing events. It is extremely important to show through experimental/FISH validations that these events are not computational artifacts and can be detected in the cell types.

It is not clear how (and how many?) splicing events in B cell trajectory were identified. Do you report any event that is differential in any of the B cell states as a cell with alternative splicing in B cell trajectory? And again, how these examples were chosen are they among the top genes in B cell trajectory?

Authors say that they detected many more events in cortex and also higher fraction of unannotated events in cortex, is this because cortex has been more deeply sampled compared to other tissues (Table 2)? Authors should account for sampling depth differences between cell types to see which one is really more enriched in alternative splicing events.

For Figure 9C, what is the AUC if the model is trained on one individual and used for prediction on another mouse?

Is not the higher fraction of events in 5' UTRs vs 3' UTR a result of the bias in your method? As you only consider events with shared 3' SS and not events with shared 5' sites in your analysis?

How did authors account for the coverage-dependent bias (as reported in https://elifesciences.org/articles/54603) which could cause spurious splicing bimodality in scRNA-Seq?

*Reviewer #2 (Recommendations for the authors):*

To demonstrate the significance of the approach a more completed performance evaluation, for example, using synthetic data, is recommended, as well as a comparison to alternative methods regarding biological significance.

---

## [Author Response]

Reviewer #1 (Recommendations for the authors):Authors have applied their method to two big scRNA-Seq datasets and have reported multiple biological discoveries from their computational analysis. However, the presentation and validation of the results should be improved. I elaborate on my comments below:One of my major concerns is about the evaluation and benchmarking analysis of scQuint. Authors have particularly reviewed some of the existing methods in the appendix, but they provided no comparison between the performance of scQuint and those methods. Particularly authors have mentioned on page 18 that they have previously analyzed these datasets with prior methods, but they did not provide any comparison between their findings and those by other methods. In its current form, it is extremely difficult to judge the sensitivity and specificity of scQuint or whether it is a new contribution to the field. Thus, the paper's contribution is to run a standard, published analysis on a single cell dataset. There is no functional or experimental validation to support or refute the findings. Further, there is no computational validation, in terms of testing whether the predictions in this dataset hold in other data.

Thank you for raising these concerns. We agree that a more rigorous evaluation of scQuint will improve the manuscript. Unfortunately, it is difficult to formally compare all of the methods we discuss because of differences in their requirements and output. For instance, nearly all methods either require annotations or don’t have two-sample tests for differential transcript usage proportions (DTU); neither of these conditions apply to scQuint. We hence reported our qualitative finding that embeddings based on the quantifications of various methods were hampered by technical biases on these datasets (Figure 1 and its two associated supplementary figures), which led us to have doubts about their performance.

We now introduce an *in silico* benchmarking procedure to demonstrate scQuint’s performance compared with other methods where such a comparison is feasible (Figure 4 in the section Differential splicing hypothesis testing with Generalized Linear Model on page 6). We had intended to include SCATS and DESJ, but were forced to exclude them due to difficulties getting their software to run (the same issues have been noted by other users on Github). To summarize, we found scQuint was comparable to the top methods in DTU testing when coverage artifacts were absent and vastly outperformed them when artifacts were present.

Per your later suggestion, we now look across different mice to help support the reliability of our findings (discussed in response to comments 4 and 6). Experimental validation, while ideal, has generally not been undertaken by relevant methods (with a couple of exceptions) and would require a time horizon and scope far beyond our intent for this work. We have begun one such collaboration, however. (See also the response to comment 9)

One limitation of the current method is more statistical tests which could lower the statistical power due to multiple hypothesis testing issue, as it needs to perform a separate test for each pair of gene/cell type compared to a test for each gene that some other methods need for finding "genes" with cell-type-specific splicing.

We understand such a test can be advantageous in certain situations. Nevertheless, we decided to test each cell type against the rest, as is the usual approach in the field (like, for example, in the tutorials for the popular single-cell analysis frameworks Seurat and Scanpy).

Authors have used the same model as in leafcutter for their analysis. However, they claim that they are getting better p-value and clustering results compared to leafcutter. It is not clear why their method should perform better than leafcutter.

Thank you for pointing out this ambiguity. We adapted several aspects of LeafCutter’s approach to make it more suitable for the scRNA-seq context, and this is the source of differences in accuracy and scalability. We now include a short description of these modifications in the Methods section and refer to these changes in the appropriate locations of the Introduction and Results. Specifically, we use a modified intron quantification (LeafCutter groups introns that share 5’ donor or 3’ acceptor sites while scQuint groups those sharing 3’ acceptor sites). Moreover, LeafCutter employs a vector of concentration parameters where entries correspond to introns whereas scQuint utilizes a single concentration parameter for each intron group. Finally, the optimization procedure of scQuint was implemented separately and makes a number of different choices, notably adopting a different initialization strategy and using different hyperparameters. The aggregate effect of these changes improves the robustness, estimability, memory requirement, and speed of the quantification and testing procedure when applied to scRNA-seq reads. See Figure 2—figure supplement 2.

One of the major advantages of the tabula muris dataset is that it contains data from multiple mouse individuals (i.e., biological replicates), which can be leveraged to show the reproducibility of the biological findings across biological replicates. However, authors did not take advantage of this in presenting their results. I highly recommend that authors show that their results can be replicated across mouse individuals, by visualizing their results as stratified by donor ID. Reproducibility is important for distinguishing between a real reproducible biological signal and a biological/technical noise particularly for the unannotated splicing events as they might be a product of splicing noise.

Absolutely, thank you for this great suggestion. We went through many genes, visualizing the results across donors and found that the splicing patterns were well-conserved. We now include supplementary figures demonstrating this for several of the events we highlighted in the manuscript (Figure 6—figure supplements 2,3,4; Figure 8—figure supplements 1,2; Figure 9—figure supplements 2,3; Figure 11—figure supplements 2,3,4). We further built a cell type classifier with no overlap in donors between the training and test sets to see if cell-type-specific patterns were preserved across mice, finding that the classifier performed just as well as when all mice were used in both the training and test sets. The updated panels Figure 10d/e now show these results.

Authors have applied their method to only SS2 and not to any 10x data. I believe that the tabula muris dataset contains 10x data as well. While I agree that 10x is more challenging than SS2 for splicing analysis, it is still a valuable resource for splicing analysis as it has higher throughput compared to SS2 and can better capture rare cell types. I recommend that authors comment on the applicability of their method to 10x in the paper and, if their method is applicable, show how their current results compare to the results based on 10x data.

This is indeed an important topic to discuss. We have added the text below to the manuscript to address your comment. Moreover, while it is time consuming to process and analyze all the 10x data from Tabula Muris, we did try a pilot analysis on a smaller, related 10x dataset, finding rather few alternative introns. For now, we think this is the safest advice.

“We do not recommend using scQuint to analyze alternative splicing in 10x Genomics Chromium data given its strong 3' transcript bias and evidence suggesting these data can detect about half the number of junctions as Smart-seq2 (Wang et al. 2021). While this imposes a fundamental limit on the number of isoforms that can be distinguished, several approaches for differential transcript usage in 10x data have been developed: Sierra (Patrick et al. 2020), SpliZ (Olivieri et al. 2021), and a kallisto-based approach which could be adapted for this task (Ntranos et al. 2019). While a systematic benchmark is missing, we expect alternative intron quantification to be sub-optimal in this setting.”

Since the paper is about analyzing splicing in single cells, I think it is extremely valuable to show the variation at the "single-cell level" (rather than pseudobulked cell-type level) via box or violin plots. This is extremely important as it is not clear from the current plots (e.g., figure 5 c,d,e or figure 8c,d) that the splicing event was observed in how many cells in each cell type and what is the range of read counts per single cell in each cell type. As I mentioned earlier it is extremely difficult to judge the reproducibility and single-cell variation of the visualized splicing events in figures as the data is aggregated across all cells within the cell type from separate donors.

We want to emphasize that all analyses (with the exception of the splice factor regression model in Figure 11) use the single cell data directly and are not pseudobulked. We now make this more explicit in the manuscript. The new supplementary figures (Figure 6—figure supplements 2,3,4; Figure 8—figure supplements 1,2; Figure 9—figure supplements 2,3; Figure 11—figure supplements 2,3,4) to which we alluded in comment 4 response show the distribution of introns across donors for these particular events and are broadly representative of the general trend of consistency we see across genes. These display the strong cell-type specificity and consistency across individuals. We did try box and violin plots, but given the relatively binary nature of the data within cell types, they didn’t yield particularly informative visualizations.

Authors mention that there is little overlap between differentially expressed and differentially spliced genes but on the other hand they say that the clusters based on splicing and expression latent space are highly consistent with each other. I think they should comment on why this is possible, is it because the same cluster has different markers in each space. If so, is it possible to highlight a few clusters and show their marker genes based on splicing and expression changes?

Thank you for raising this question, as it is an interesting and important one which gets at the biological phenomena at play. Indeed, we found that the markers are generally distinct in the expression and splicing spaces despite both data modalities yielding highly consistent clusterings. Figures 6b and 9a-b show that the pattern of alternative splicing across cell types generally do not correlate with the expression of corresponding genes across the same cell types. Table 2 also shows the extremely narrow overlap of the most prominent markers in these two spaces for 5 different tissues. We have also added Figure 6—figure supplement 1 to display the expression marker genes for the cortex cell types. Inspection reveals very little overlap with the splicing markers.

On page 9, authors say that they detected thousands of cell-type-specific events; however, they do not provide more specifics about these events? How many events exactly? Across how many distinct genes (also what fraction of genes, and is this fraction with previous studies?)? And distinct cell types? Also, it is not clear how the examples in figure 5 were chosen? Are they among the top genes? What are the top genes? Are they genes known to have cell-type-specific splicing?

Figure 10 contains information about the number of differential splicing events and differentially spliced genes across different cell types in both the Cortex and Tabula Muris datasets. We have inserted a parenthetical remark in the sentence you mentioned to direct readers to the latter section that contains this figure and a discussion of the information it contains.

The splicing events in Figure 6 (the former figure 5) panel b were chosen after ranking by inferred effect size with some manual curation to ensure cell-type specificity. Genes for panels c-e (and for subsequent Sashimi plots) were selected from our pool of cell-type-specific novel events based on the clarity of alternative splicing/transcription, and genes were prioritized if there was prior knowledge of roles in relevant biological processes.

The paper lacks any experimental validation on the discovered splicing events. It is extremely important to show through experimental/FISH validations that these events are not computational artifacts and can be detected in the cell types.

We agree that experimental validation is a vital avenue to pursue, and we have begun a collaboration to investigate some of the computationally inferred splicing differences we found in neuronal cells. However, for reasons of the necessary time and our intended scope, we do not feel that it is appropriate to include that as part of this manuscript. We also note that the large majority of computational methods papers for splicing analysis do not perform any experimental validation (we found only 2 out of 13 did so). Our hope with this work is to help generate reasonable hypotheses for follow-up experimentation given the expense of such an endeavor.

It is not clear how (and how many?) splicing events in B cell trajectory were identified. Do you report any event that is differential in any of the B cell states as a cell with alternative splicing in B cell trajectory? And again, how these examples were chosen are they among the top genes in B cell trajectory?

We have inserted panel b of Figure 8 (formerly Figure 7) to explicitly give the number of identified splicing events for each B cell stage. Examples were chosen by selecting genes we identified as containing differential splicing/transcription start events across B cell stages which also have previously identified roles (based on expression) in B cell development. This allowed a comparison of the progression of splicing and expression in genes where both processes seem to play a role in development.

Authors say that they detected many more events in cortex and also higher fraction of unannotated events in cortex, is this because cortex has been more deeply sampled compared to other tissues (Table 2)? Authors should account for sampling depth differences between cell types to see which one is really more enriched in alternative splicing events.

We have updated the relevant section so that it provides readers better context for our results. In particular, we note that the higher rate of unannotated events in the cortex is consistent with previous observations. Due to the one-vs-all testing procedure, the results are affected by the sequencing depths and number of cell types across many tissues and cell types. This makes a comprehensive evaluation difficult, but we now mention that these may complicate a direct interpretation of the number of identified events. The newly added items are underlined.

“We found many more cell-type-specific differential splicing events in the cortex than in the marrow, as expected (Yeo et al. 2004.), as well as a higher proportion of events involving novel junctions, which can reach 30% (Figure 10a). Differences in proportion of novel junctions should be interpreted with care, however, since they can be affected by sequencing depth and number of cells, both of which vary between the two tissues. Very similar patterns are seen when grouping differential splicing events that occur in the same gene (Figure 10b).”

For Figure 9C, what is the AUC if the model is trained on one individual and used for prediction on another mouse?

Figure 10d now shows the AUC when the classifier is trained and tested on different sets of mice. The values are extremely similar to when there was no partition of individuals between the training and testing data.

Is not the higher fraction of events in 5' UTRs vs 3' UTR a result of the bias in your method? As you only consider events with shared 3' SS and not events with shared 5' sites in your analysis?

We now include a note that our method of intron quantification influences the 5’ vs 3’ ratio we observe:

"Most differential splicing events that we detected with alternative introns fall in the coding portion of the gene, with high proportions in the 5' UTR (Figure 10c). This is a property of our quantification approach and does not reflect the total number of alternative splicing events in different gene regions; still, the relative proportion can be compared across tissues.”

How did authors account for the coverage-dependent bias (as reported in https://elifesciences.org/articles/54603) which could cause spurious splicing bimodality in scRNA-Seq?

We also encountered this issue while working with these datasets, and it was an important consideration to select a model robust to such technical noise. The Dirichlet-Multinomial model we chose is able to handle this bimodality by fitting a concentration parameter close to zero. To clarify this for readers, we have inserted the following sentence:

“For example, the often encountered “binary'' splicing (Najar et al. 2020) can be modeled by fitting a concentration parameter close to zero.”

Reviewer #2 (Recommendations for the authors):To demonstrate the significance of the approach a more completed performance evaluation, for example, using synthetic data, is recommended, as well as a comparison to alternative methods regarding biological significance.

We thank the reviewer for their careful reading of our manuscript and for noting its strengths and weaknesses. We have revised the paper with a new computational benchmarking section and a better discussion of the biological novelty and insights yielded by our method in this analysis.